# Multi-Objective Online Learning

**Jiyan Jiang**[*]
Tsinghua University
Beijing, China
scjjy95@outlook.com

**Wenpeng Zhang**[*][†]
Ant Group
Beijing, China
zhangwenpeng0@gmail.com

**Shiji Zhou**
Tsinghua University
Beijing, China
zhoushiji00@gmail.com

**Lihong Gu, Xiaodong Zeng**
Ant Group
Hangzhou, China
{lihong.glh,xiaodong.zxd}@antgroup.com

**Wenwu Zhu**[†]
Tsinghua University
Beijing, China
wwzhu@tsinghua.edu.cn

## Abstract

This paper presents a systematic study of multi-objective online learning. We first formulate the framework of Multi-Objective Online Convex Optimization, which encompasses a novel multi-objective regret. This regret is built upon a sequence-wise extension of the commonly used discrepancy metric Pareto suboptimality gap in zero-order multi-objective bandits. We then derive an equivalent form of the regret, making it amenable to be optimized via first-order iterative methods. To motivate the algorithm design, we give an explicit example in which equipping OMD with the vanilla min-norm solver for gradient composition will incur a linear regret, which shows that merely regularizing the iterates, as in single-objective online learning, is not enough to guarantee sublinear regrets in the multi-objective setting. To resolve this issue, we propose a novel min-regularized-norm solver that regularizes the composite weights. Combining min-regularized-norm with OMD results in the Doubly Regularized Online Mirror Multiple Descent algorithm. We further derive the multi-objective regret bound for the proposed algorithm, which matches the optimal bound in the single-objective setting. Extensive experiments on several real-world datasets verify the effectiveness of the proposed algorithm.

## 1 Introduction

Traditional optimization methods for machine learning are usually designed to optimize a single objective. However, in many real-world applications, we are often required to optimize multiple correlated objectives concurrently. For example, in autonomous driving (Huang et al., 2019; Lu et al., 2019b), self-driving vehicles need to solve multiple tasks such as self-localization and object identification at the same time. In online advertising (Ma et al., 2018a;b), advertising systems need to decide on the exposure of items to different users to maximize both the Click-Through Rate (CTR) and the Post-Click Conversion Rate (CVR). In most multi-objective scenarios, the objectives may conflict with each other (Kendall et al., 2018). Hence, there may not exist any single solution that can optimize all the objectives simultaneously. For example, merely optimizing CTR or CVR will degrade the performance of the other (Ma et al., 2018a;b).

Multi-objective optimization (MOO) (Marler & Arora, 2004; Deb, 2014) is concerned with optimizing multiple conflicting objectives simultaneously. It seeks Pareto optimality, where no single objective can be improved without hurting the performance of others. Many different methods for MOO have been proposed, including evolutionary methods (Murata et al., 1995; Zitzler & Thiele, 1999), scalarization methods (Fliege & Svaiter, 2000), and gradient-based iterative methods (Désidéri, 2012). Recently, the Multiple Gradient Descent Algorithm (MGDA) and its variants have been introduced to the training of multi-task deep neural networks and achieved great empirical success (Sener & Koltun, 2018), making them regain a significant amount of research interest (Lin et al., 2019; Yu et al., 2020; Liu et al., 2021). These methods compute a composite gradient based on

---

[*]Equal contributions.    [†]Corresponding author.

the gradient information of all the individual objectives and then apply the composite gradient to update the model parameters. The composite weights are determined by a min-norm solver (Désidéri, 2012) which yields a common descent direction of all the objectives.

However, compared to the increasingly wide application prospect, the gradient-based iterative algorithms are relatively understudied, especially in the online learning setting. Multi-objective online learning is of essential importance for reasons in two folds. First, due to the data explosion in many real-world scenarios such as web applications, making in-time predictions requires performing online learning. Second, the theoretical investigation of multi-objective online learning will lay a solid foundation for the design of new optimizers for multi-task deep learning. This is analogous to the single-objective setting, where nearly all the optimizers for training DNNs are initially analyzed in the online setting, such as AdaGrad (Duchi et al., 2011), Adam (Kingma & Ba, 2015), and AMS-Grad (Reddi et al., 2018).

In this paper, we give a systematic study of multi-objective online learning. To begin with, we formulate the framework of Multi-Objective Online Convex Optimization (MO-OCO). One major challenge in deriving MO-OCO is the lack of a proper regret definition. In the multi-objective setting, in general, no single decision can optimize all the objectives simultaneously. Thus, to devise the multi-objective regret, we need to first extend the single fixed comparator used in the single-objective regret, i.e., the fixed optimal decision, to the entire Pareto optimal set. Then we need an appropriate discrepancy metric to evaluate the gap between vector-valued losses. Intuitively, the Pareto suboptimality gap (PSG) metric, which is frequently used in zero-order multi-objective bandits (Turgay et al., 2018; Lu et al., 2019a), is a very promising candidate. PSG can yield scalarized measurements from any vector-valued loss to a given comparator set. However, we find that vanilla PSG is unsuitable for our setting since it always yields non-negative values and may be too loose. In a concrete example, we show that the naive PSG-based regret $R_{\mathrm{I}}(T)$ can even be linear w.r.t. $T$ when the decisions are already optimal, which disqualifies it as a regret metric. To overcome the failure of vanilla PSG, we propose its sequence-wise variant termed S-PSG, which measures the suboptimality of the whole decision sequence to the Pareto optimal set of the cumulative loss function. Optimizing the resulting regret $R_{\mathrm{II}}(T)$ will drive the cumulative loss to approach the Pareto front. However, as a zero-order metric motivated geometrically, designing appropriate first-order algorithms to directly optimize it is too difficult. To resolve the issue, we derive a more intuitive equivalent form of $R_{\mathrm{II}}(T)$ via a highly non-trivial transformation.

Based on the MO-OCO framework, we develop a novel multi-objective online algorithm termed Doubly Regularized Online Mirror Multiple Descent. The key module of the algorithm is the gradient composition scheme, which calculates a composite gradient in the form of a convex combination of the gradients of all objectives. Intuitively, the most direct way to determine the composite weights is to apply the min-norm solver (Désidéri, 2012) commonly used in offline multi-objective optimization. However, directly applying min-norm is not workable in the online setting. Specifically, the composite weights in min-norm are merely determined by the gradients at the current round. In the online setting, since the gradients are adversarial, they may result in undesired composite weights, which further produce a composite gradient that reversely optimizes the loss. To rigorously verify this point, we give an example where equipping OMD with vanilla min-norm incurs a linear regret, showing that only regularizing the iterate, as in OMD, is not enough to guarantee sublinear regrets in our setting. To fix the issue, we devise a novel min-regularized-norm solver with an explicit regularization on composite weights. Equipping it with OMD results in our proposed algorithm.

In theory, we derive a regret bound of $O(\sqrt{T})$ for DR-OMMD, which matches the optimal bound in the single-objective setting (Hazan et al., 2016) and is tight w.r.t. the number of objectives. Our analysis also shows that DR-OMMD attains a smaller regret bound than that of linearization with fixed composite weights. We show that, in the two-objective setting with linear losses, the margin between the regret bounds depends on the difference between the composite weights yielded by the two algorithms and the difference between the gradients of the two underlying objectives.

To evaluate the effectiveness of DR-OMMD, we conduct extensive experiments on several large-scale real-world datasets. We first realize adaptive regularization via multi-objective optimization, and find that adaptive regularization with DR-OMMD significantly outperforms fixed regularization with linearization, which verifies the effectiveness of DR-OMMD over linearization in the convex setting. Then we apply DR-OMMD to deep online multi-task learning. The results show that DR-OMMD is also effective in the non-convex setting.

## 2 PRELIMINARIES

In this section, we briefly review the necessary background knowledge of two related fields.

### 2.1 MULTI-OBJECTIVE OPTIMIZATION

**Multiple-objective optimization (MOO)** is concerned with solving the problems of optimizing multiple objectives simultaneously (Fliege & Svaiter, 2000; Deb, 2014). In general, since different objectives may conflict with each other, there is no single solution that can optimize all the objectives at the same time, hence the conventional concept of optimality used in the single-objective setting is no longer suitable. Instead, MOO seeks to achieve Pareto optimality. In the following, we give the relevant definitions more formally. We use a vector-valued loss $F = (f^1, \ldots, f^m)$ to denote the objectives, where $m \geq 2$ and $f^i : \mathcal{X} \to \mathbb{R}$, $i \in \{1, \ldots, m\}$, $\mathcal{X} \subset \mathbb{R}$, is the $i$-th loss function.

**Definition 1 (Pareto optimality). (a)** For any two solutions $\boldsymbol{x}, \boldsymbol{x}' \in \mathcal{X}$, we say that $\boldsymbol{x}$ dominates $\boldsymbol{x}'$, denoted as $\boldsymbol{x} \prec \boldsymbol{x}'$ or $\boldsymbol{x}' \succ \boldsymbol{x}$, if $f^i(\boldsymbol{x}) \leq f^i(\boldsymbol{x}')$ for all $i$, and there exists one $i$ such that $f^i(\boldsymbol{x}) < f^i(\boldsymbol{x}')$; otherwise, we say that $\boldsymbol{x}$ does not dominate $\boldsymbol{x}'$, denoted as $\boldsymbol{x} \not\prec \boldsymbol{x}'$ or $\boldsymbol{x}' \not\succ \boldsymbol{x}$.
**(b)** A solution $\boldsymbol{x}^* \in \mathcal{X}$ is called Pareto optimal if it is not dominated by any other solution in $\mathcal{X}$.

Note that there may exist multiple Pareto optimal solutions. For example, it is easy to show that the optimizer of any single objective, i.e., $\boldsymbol{x}_i^* \in \arg\min_{\boldsymbol{x} \in \mathcal{X}} f^i(\boldsymbol{x})$, $i \in \{1, \ldots, m\}$, is Pareto optimal. Different Pareto optimal solutions reflect different trade-offs among the objectives (Lin et al., 2019).

**Definition 2 (Pareto front). (a)** All Pareto optimal solutions form the Pareto set $\mathcal{P}_{\mathcal{X}}(F)$.
**(b)** The image of $\mathcal{P}_{\mathcal{X}}(F)$ constitutes the Pareto front, denoted as $\mathcal{P}(H) = \{F(\boldsymbol{x}) \mid \boldsymbol{x} \in \mathcal{P}_{\mathcal{X}}(F)\}$.

Now that we have established the notion of optimality in MOO, we proceed to introduce the metrics that measure the discrepancy of an arbitrary solution $\boldsymbol{x} \in \mathcal{X}$ from being optimal. Recall that, in the single-objective setting with merely one loss function $f : \mathcal{Z} \to \mathbb{R}$, for any $\boldsymbol{z} \in \mathcal{Z}$, the loss difference $f(\boldsymbol{z}) - \min_{\boldsymbol{z}'' \in \mathcal{Z}} f(\boldsymbol{z}'')$ is directly qualified for the discrepancy measure. However, in MOO with more than one loss, for any $\boldsymbol{x} \in \mathcal{X}$, the loss difference $F(\boldsymbol{x}) - F(\boldsymbol{x}'')$, where $\boldsymbol{x}'' \in \mathcal{P}_{\mathcal{X}}(F)$, is a vector. Intuitively, the desired discrepancy metric shall scalarize the vector-valued loss difference and yield 0 for any Pareto optimal solution. In general, in MOO, there are two commonly used discrepancy metrics, i.e., Pareto suboptimality gap (PSG) (Turgay et al., 2018) and Hypervolume (HV) (Bradstreet, 2011). As HV is a complex volume-based metric, it is more difficult to optimize via gradient-based algorithms (Zhang & Golovin, 2020). Hence in this paper, we adopt PSG, which has already been extensively used in multi-objective bandits (Turgay et al., 2018; Lu et al., 2019a).

**Definition 3 (Pareto suboptimality gap[1]).** For any $\boldsymbol{x} \in \mathcal{X}$, the Pareto suboptimality gap to a given comparator set $\mathcal{Z} \subset \mathcal{X}$, denoted as $\Delta(\boldsymbol{x}; \mathcal{Z}, F)$, is defined as the minimal scalar $\epsilon \geq 0$ that needs to be subtracted from all entries of $F(\boldsymbol{x})$, such that $F(\boldsymbol{x}) - \epsilon \mathbf{1}$ is not dominated by any point in $\mathcal{Z}$, where $\mathbf{1}$ denotes the all-one vector in $\mathbb{R}^m$, i.e.,

$$\Delta(\boldsymbol{x}; \mathcal{Z}, F) = \inf_{\epsilon \geq 0} \epsilon, \text{ s.t. } \forall \boldsymbol{x}'' \in \mathcal{Z}, \ \exists i \in \{1, \ldots, m\}, \ f^i(\boldsymbol{x}) - \epsilon < f^i(\boldsymbol{x}'').$$

Clearly, PSG is a distance-based discrepancy metric motivated from a purely geometric viewpoint. In practice, the comparator set $\mathcal{Z}$ is often set to be the Pareto set $\mathcal{X}^* = \mathcal{P}_{\mathcal{X}}(F)$ (Turgay et al., 2018); therein for any $\boldsymbol{x} \in \mathcal{K}$, its PSG is always non-negative and equals zero if and only if $\boldsymbol{x} \in \mathcal{P}_{\mathcal{X}}(F)$.

**Multiple Gradient Descent Algorithm (MGDA)** is an offline first-order MOO algorithm (Fliege & Svaiter, 2000; Désidéri, 2012). At each iteration $l \in \{1, \ldots, L\}$ ($L$ is the number of iterations), it first computes the gradient $\nabla f^i(\boldsymbol{x}_l)$ of each objective, then derives the composite gradient $\boldsymbol{g}_l^{comp} = \sum_{i=1}^m \lambda_l^i \nabla f^i(\boldsymbol{x}_l)$ as a convex combination of these gradients, and finally applies $\boldsymbol{g}_l^{comp}$ to execute a gradient descent step to update the decision, i.e., $\boldsymbol{x}_{l+1} = \boldsymbol{x}_l - \eta \boldsymbol{g}_l^{comp}$ ($\eta$ is the step size). The core part of MGDA is the module that determines the composite weights $\boldsymbol{\lambda}_l = (\lambda_l^1, \ldots, \lambda_l^m)$, given by

$$\boldsymbol{\lambda}_l = \arg\min_{\boldsymbol{\lambda}_l \in \mathcal{S}_m} \| \sum_{i=1}^m \lambda_l^i \nabla f^i(\boldsymbol{x}_l) \|_2^2,$$

where $\mathcal{S}_m = \{\boldsymbol{\lambda} \in \mathbb{R}^m \mid \sum_{i=1}^m \lambda^i = 1, \lambda^i \geq 0, i \in \{1, \ldots, m\}\}$ is the probabilistic simplex in $\mathbb{R}^m$. This is a min-norm solver, which finds the weights in the simplex that yield the minimum L2-norm of the composite gradient. Thus MGDA is also called the *min-norm* method. Previous works

---

[1]Our definition looks a bit different from (Turgay et al., 2018). In Appendix B, we show they are equivalent.

(Désidéri, 2012; Sener & Koltun, 2018) showed that when all $f^i$ are convex functions, MGDA is guaranteed to decrease all the objectives simultaneously until it reaches a Pareto optimal decision.

## 2.2 Online Convex Optimization

**Online Convex Optimization (OCO)** (Zinkevich, 2003; Hazan et al., 2016) is the most commonly adopted framework for designing online learning algorithms. It can be viewed as a structured repeated game between a learner and an adversary. At each round $t \in \{1, \ldots, T\}$, the learner is required to generate a decision $\boldsymbol{x}_t$ from a convex compact set $\mathcal{X} \subset \mathbb{R}^n$. Then the adversary replies the learner with a convex function $f_t : \mathcal{X} \to \mathbb{R}$ and the learner suffers the loss $f_t(\boldsymbol{x}_t)$. The goal of the learner is to minimize the regret with respect to the best fixed decision in hindsight, i.e.,

$$R(T) = \sum\nolimits_{t=1}^{T} f_t(\boldsymbol{x}_t) - \min_{\boldsymbol{x}^* \in \mathcal{X}} \sum\nolimits_{t=1}^{T} f_t(\boldsymbol{x}^*).$$

A meaningful regret is required to be sublinear in $T$, i.e., $\lim_{T \to \infty} R(T)/T = 0$, which implies that when $T$ is large enough, the learner can perform as well as the best fixed decision in hindsight.

**Online Mirror Descent (OMD)** (Hazan et al., 2016) is a classic first-order online learning algorithm. At each round $t \in \{1, \ldots, T\}$, OMD yields its decision via

$$\boldsymbol{x}_{t+1} = \arg\min_{\boldsymbol{x} \in \mathcal{X}} \eta \langle \nabla f_t(\boldsymbol{x}_t), \boldsymbol{x} \rangle + B_R(\boldsymbol{x}, \boldsymbol{x}_t),$$

where $\eta$ is the step size, $R : \mathcal{X} \to \mathbb{R}$ is the regularization function, and $B_R(\boldsymbol{x}, \boldsymbol{x}') = R(\boldsymbol{x}) - R(\boldsymbol{x}') - \langle \nabla R(\boldsymbol{x}'), \boldsymbol{x} - \boldsymbol{x}' \rangle$ is the Bregman divergence induced by $R$. As a meta-algorithm, by instantiating different regularization functions, OMD can induce two important algorithms, i.e., Online Gradient Descent (Zinkevich, 2003) and Online Exponentiated Gradient (Hazan et al., 2016).

## 3 Multi-Objective Online Convex Optimization

In this section, we formally formulate the MO-OCO framework.

**Framework overview.** Analogously to single-objective OCO, MO-OCO can be viewed as a repeated game between an online learner and the adversarial environment. The main difference is that in MO-OCO, the feedback is vector-valued. The general framework of MO-OCO is given as follows. At each round $t \in \{1, \ldots, T\}$, the learner generates a decision $\boldsymbol{x}_t$ from a given convex compact decision set $\mathcal{X} \subset \mathbb{R}^n$. Then the adversary replies the decision with a vector-valued loss function $F_t : \mathcal{X} \to \mathbb{R}^m$, whose $i$-th component $f_t^i : \mathcal{X} \to \mathbb{R}$ is a convex function corresponding to the $i$-th objective, and the learner suffers the vector-valued loss $F_t(\boldsymbol{x}_t)$. The goal of the learner is to generate a sequence of decisions $\{\boldsymbol{x}_t\}_{t=1}^{T}$ to minimize a certain kind of multi-objective regret.

The remaining work in framework formulation is to give an appropriate regret definition, which is the most challenging part. Recall that the single-objective regret $R(T) = \sum_{t=1}^{T} f_t(\boldsymbol{x}_t) - \sum_{t=1}^{T} f_t(\boldsymbol{x}^*)$ is defined as the difference between the cumulative loss of the actual decisions $\{\boldsymbol{x}_t\}_{t=1}^{T}$ and that of the fixed optimal decision in hindsight $\boldsymbol{x}^* \in \arg\min_{\boldsymbol{x} \in \mathcal{X}} \sum_{t=1}^{T} f_t(\boldsymbol{x})$. When defining the multi-objective analogy to $R(T)$, we encounter two issues. First, in the multi-objective setting, no single decision can optimize all the objectives simultaneously in general, hence we cannot compare the cumulative loss with that of any single decision. Instead, we use the the Pareto optimal set $\mathcal{X}^*$ of the cumulative loss function $\sum_{t=1}^{T} F_t$, i.e., $\mathcal{X}^* = \mathcal{P}_X(\sum_{t=1}^{T} F_t)$, which naturally aligns with the optimality concept in MOO. Second, to compare $\{\boldsymbol{x}_t\}_{t=1}^{T}$ and $\mathcal{X}^*$ in the loss space, we need a discrepancy metric to measure the gap between vector losses. Intuitively, we can adopt the commonly used PSG metric (Turgay et al., 2018). But we find that vanilla PSG is not appropriate for OCO, which is largely different from the bandits setting. We explicate the reason in the following.

### 3.1 The Naive Regret based on Vanilla PSG Fails in MO-OCO

By definition, at each round $t$, the difference between the decision $\boldsymbol{x}_t$ and the Pareto optimal set can be evaluated by PSG $\Delta(\boldsymbol{x}_t; \mathcal{X}^*, F_t)$. Naturally, we can formulate the multi-objective regret by accumulating $\Delta(\boldsymbol{x}_t; \mathcal{X}^*, F_t)$ over all rounds, i.e.,

$$R_{\mathrm{I}}(T) := \sum\nolimits_{t=1}^{T} \Delta(\boldsymbol{x}_t; \mathcal{X}^*, F_t).$$

Recall that the single-objective regret can also expressed as $R(T) = \sum_{t=1}^{T}(f_t(\boldsymbol{x}_t) - f_t(\boldsymbol{x}^*))$. Hence, $R_{\mathrm{I}}(T)$ essentially extends the scalar discrepancy $f_t(\boldsymbol{x}_t) - f_t(\boldsymbol{x}^*)$ to the PSG metric $\Delta(\boldsymbol{x}_t; \mathcal{X}^*, F_t)$. However, these two discrepancy metrics have a major difference, i.e., $f_t(\boldsymbol{x}_t) - f_t(\boldsymbol{x}^*)$ can be negative, whereas $\Delta(\boldsymbol{x}_t; \mathcal{X}^*, F_t)$ is always non-negative. In previous bandits settings (Turgay et al., 2018), the discrepancy is intrinsically non-negative, since the comparator set is exactly the Pareto optimal set of the evaluated loss function. However, the non-negative property of PSG can be problematic in our setting, where the comparator set $\mathcal{X}^*$ is the Pareto set of the cumulative loss function, rather than the instantaneous loss $F_t$ that is used for evaluation. Specifically, at some round $t$, the decision $x_t$ may Pareto dominate all points in $\mathcal{X}^*$ w.r.t. $F_t$, which corresponds to the single-objective setting where it is possible that $f_t(\boldsymbol{x}_t) < f_t(\boldsymbol{x}^*)$ at some specific round. In this case, we would expect the discrepancy metric at this round to be negative. However, PSG can only yield 0 in this case, making the regret much looser than we expect. In the following, we provide an example in which the naive regret $R_{\mathrm{I}}(T)$ is **linear** w.r.t. $T$ even when the decisions $x_t$ are already optimal.

**Problem instance.** Set $\mathcal{X} = [-2, 2]$. Let the loss function be identical among all objectives, i.e., $f_t^1(x) = \dots = f_t^m(x)$, and alternate between $x$ and $-x$. Suppose the time horizon $T$ is an even number, then the Pareto optimal set $\mathcal{X}^* = \mathcal{X}$. Now consider the decisions $x_t = 1, t \in \{1, \dots, T\}$. In this case, it can easily be checked that the single-objective regret of each objective is zero, indicating that these decisions are optimal for each objective. To calculate $R_{\mathrm{I}}(T)$, notice that when all the objectives are identical, PSG reduces to $\Delta(x_t; \mathcal{X}^*, f_t^1) = \sup_{x^* \in \mathcal{X}} \max\{f_t^1(x_t) - f_t^1(x^*), 0\}$ at each round $t$. Hence, in this case we have $R_{\mathrm{I}}(T) = \sum_{1 \le k \le T/2}(\sup_{x^* \in [-2, 2]} \max\{1 - x^*, 0\} + \sup_{x^* \in [-2, 2]} \max\{x^* - 1, 0\}) = 3T$, which is linear w.r.t. $T$. Therefore, $R_{\mathrm{I}}(T)$ is too loose to measure the suboptimality of decisions, which is **unqualified as a regret metric**.

## 3.2 THE ALTERNATIVE REGRET BASED ON SEQUENCE-WISE PSG

In light of the failure of the naive regret, we need to modify the discrepancy metric in our setting. Recall that the single-objective regret can be interpreted as the gap between the actual cumulative loss $\sum_{t=1}^{T} f_t(\boldsymbol{x}_t)$ and its optimal value $\min_{\boldsymbol{x} \in \mathcal{X}} \sum_{t=1}^{T} f_t(\boldsymbol{x})$. In analogy, we can measure the gap between $\sum_{t=1}^{T} F_t(\boldsymbol{x}_t)$ and the Pareto front $\mathcal{P}^* = \mathcal{P}_{\mathcal{X}}(\sum_{t=1}^{T} F_t)$. However, vanilla PSG is a pointwise metric, i.e., it can only measure the suboptimality of a decision point. To evaluate the decision sequence $\{\boldsymbol{x}_t\}_{t=1}^{T}$, we modify its definition and propose a sequence-wise variant of PSG.

**Definition 4 (Sequence-wise PSG).** For any decision sequence $\{\boldsymbol{x}_t\}_{t=1}^{T}$, the sequence-wise PSG (S-PSG) to a given comparator set[2] $\mathcal{X}^*$ w.r.t. the loss sequence $\{F_t\}_{t=1}^{T}$ is defined as

$$\Delta(\{\boldsymbol{x}_t\}_{t=1}^{T}; \mathcal{X}^*, \{F_t\}_{t=1}^{T}) = \inf_{\epsilon \ge 0} \epsilon, \text{ s.t. } \forall \boldsymbol{x}'' \in \mathcal{X}^*, \exists i \in \{1, \dots, m\}, \sum_{t=1}^{T} f_t^i(\boldsymbol{x}_t) - \epsilon < \sum_{t=1}^{T} f_t^i(\boldsymbol{x}'').$$

Since $\mathcal{X}^*$ is the Pareto set of $\sum_{t=1}^{T} F_t$, S-PSG measures the discrepancy from the cumulative loss of the decision sequence to the Pareto front $\mathcal{P}^*$. Now the regret can be directly given as

$$R_{\mathrm{II}}(T) := \Delta(\{\boldsymbol{x}_t\}_{t=1}^{T}; \mathcal{X}^*, \{F_t\}_{t=1}^{T}).$$

$R_{\mathrm{II}}(T)$ has a clear physical meaning that optimizing it will impose the cumulative loss to be close to the Pareto front $\mathcal{P}^*$. However, since PSG (or S-PSG) is a zero-order metric motivated in a purely geometric sense, i.e., its calculation needs to solve a constrained optimization problem with an unknown boundary $\{F_t(\boldsymbol{x}'') \mid \boldsymbol{x}'' \in \mathcal{X}^*\}$, it is difficult to design a first-order algorithm to optimize PSG-based regrets, not to mention the analysis. To resolve this issue, we derive an equivalent form via highly non-trivial transformations, which is more intuitive than its original form.

**Proposition 1.** *The multi-objective regret $R_{\mathrm{II}}(T)$ based on S-PSG has an equivalent form, i.e.,*

$$R_{\mathrm{II}}(T) = \max \left\{ \sup_{\boldsymbol{x}^* \in \mathcal{X}^*} \inf_{\boldsymbol{\lambda}^* \in \mathcal{S}_m} \sum_{t=1}^{T} \boldsymbol{\lambda}^{*\top}(F_t(\boldsymbol{x}_t) - F_t(\boldsymbol{x}^*)), \ 0 \right\}.$$

***Remark.*** (i) The above form is closely related to the single-objective regret $R(T)$. Specifically, when $m = 1$, we can prove that $R_{\mathrm{II}}(T) = \max\{\sum_{t=1}^{T} F_t(\boldsymbol{x}_t) - \min_{\boldsymbol{x}^* \in \mathcal{X}^*} \sum_{t=1}^{T} F_t(\boldsymbol{x}^*), 0\} =$

---

[2]It is equivalent to use either $\mathcal{X}^*$ or $\mathcal{X}$ as the comparator set. See Appendix C for the detailed proof.

---

**Algorithm 1** Doubly Regularized Online Mirror Multiple Descent (**DR-OMMD**)

---

1: **Input:** Convex set $\mathcal{X}$, time horizon $T$, regularization parameter $\alpha_t$, learning rate $\eta_t$, regularization function $R$, user preference $\boldsymbol{\lambda}_0$.
2: **Initialize:** $\boldsymbol{x}_1 \in \mathcal{X}$.
3: **for** $t = 1, \dots, T$ **do**
4:    Predict $\boldsymbol{x}_t$ and receive a loss function $F_t : \mathcal{X} \to \mathbb{R}^m$.
5:    Compute the multiple gradients $\nabla F_t(\boldsymbol{x}_t) = [\nabla f_t^1(\boldsymbol{x}_t), \dots, \nabla f_t^m(\boldsymbol{x}_t)] \in \mathbb{R}^{n \times m}$.
6:    Determine the weights for the gradient composition via **min-regularized-norm**
$$\boldsymbol{\lambda}_t = \arg\min_{\boldsymbol{\lambda} \in \mathcal{S}_m} \|\nabla F_t(\boldsymbol{x}_t)\boldsymbol{\lambda}\|_2^2 + \alpha_t \|\boldsymbol{\lambda} - \boldsymbol{\lambda}_0\|_1.$$
7:    Compute the composite gradient $\boldsymbol{g}_t = \nabla F_t(\boldsymbol{x}_t)\boldsymbol{\lambda}_t$.
8:    Perform online mirror descent using $\boldsymbol{g}_t$
$$\boldsymbol{x}_{t+1} = \arg\min_{\boldsymbol{x} \in \mathcal{X}} \eta_t \langle \boldsymbol{g}_t, \boldsymbol{x} \rangle + B_R(\boldsymbol{x}, \boldsymbol{x}_t).$$
9: **end for**

---

$\max\{R(T), 0\}$. Note that in the regret analysis, we are more interested in the case of $R(T) \geq 0$ (where $R_{\mathrm{II}}(T) = R(T)$), since when $R(T) < 0$, it is naturally bounded by any sublinear regret bound. Hence, $R_{\mathrm{II}}(T)$ is essentially aligned with $R(T)$ in the single-objective setting.

(ii) At its first glance, $R_{\mathrm{II}}(T)$ can be optimized via linearization with fixed weights $\boldsymbol{\lambda}_0 \in \mathcal{S}_m$, or alternatively, optimizing a single objective $i \in \{1, ..., m\}$. We remark that this is not a problem of our regret definition, but **an intrinsic requirement** of Pareto optimality. Specifically, Pareto optimality characterizes the status where no objective can be improved without hurting others. Hence merely optimizing a single objective naturally achieves Pareto optimality. Please refer to Proposition 8 in (Emmerich & Deutz, 2018) for the rigorous proof. As a general performance metric, our regret should incorporate this special case. Later, we will design a novel algorithm based on the concept of common descent, which outperforms linearization in both theory and experiment.

## 4 Doubly Regularized Online Mirror Multiple Descent

In this section, we present the Doubly Robust Online Mirror Multiple Descent (DR-OMMD) algorithm, the protocol of which is given in Algorithm 1. At each round $t$, the learner first computes the gradient of the loss regarding each objective, then determines the composite weights of all these gradients, and finally applies the composite gradient to the online mirror descent step.

### 4.1 Vanilla Min-Norm May Incur Linear Regrets

The core module of DR-OMMD is the composition of gradients. For simplicity, denote the gradients at round $t$ in a matrix form $\nabla F_t(\boldsymbol{x}_t) = [\nabla f_t^1(\boldsymbol{x}_t), \dots, \nabla f_t^m(\boldsymbol{x}_t)] \in \mathbb{R}^{n \times m}$. Then the composite gradient is $\boldsymbol{g}_t = \nabla F_t(\boldsymbol{x}_t)\boldsymbol{\lambda}_t$, where $\boldsymbol{\lambda}_t$ is the composite weights. As illustrated in the preliminary, in the offline setting, the min-norm method (Désidéri, 2012; Sener & Koltun, 2018) is a classic method to determine the composite weights, which produces a common descent direction that can descend all the losses simultaneously. Thus, it is tempting to consider applying it to the online setting.

However, directly applying min-norm to the online setting is not workable, which may even incur linear regrets. In vanilla min-norm, the composite weights $\boldsymbol{\lambda}_t$ are determined solely by the gradients $\nabla F_t(\boldsymbol{x}_t)$ at the current round $t$, which are very sensitive to the instantaneous loss $F_t$. In the online setting, the losses at each round can be adversarially chosen, and thus the corresponding gradients can be adversarial. These adversarial gradients may result in undesired composite weights, which may further produce a composite gradient that even deteriorates the next prediction. In the following, we provide an example in which min-norm incurs a linear regret. We extend OMD (Hazan et al., 2016) to the multi-objective setting, where the composite weights are directly yielded by min-norm.

**Problem instance.** We consider a two-objective problem. The decision domain is $\mathcal{X} = \{(u, v) \mid u + v \leq \frac{1}{2}, v - u \leq \frac{1}{2}, v \geq 0\}$ and the loss function at each round is

$$F_t(\boldsymbol{x}) = \begin{cases} (\|\boldsymbol{x} - \boldsymbol{a}\|^2, \|\boldsymbol{x} - \boldsymbol{b}\|^2), & t = 2k-1, \quad k = 1, 2, ...; \\ (\|\boldsymbol{x} - \boldsymbol{b}\|^2, \|\boldsymbol{x} - \boldsymbol{c}\|^2), & t = 2k, \qquad k = 1, 2, ..., \end{cases}$$

where $\boldsymbol{a} = (-2,-1), \boldsymbol{b} = (0,1), \boldsymbol{c} = (2,-1)$. For simplicity, we first analyze the case where the total time horizon $T$ is an even number. Then we can compute the Pareto set of the cumulative loss $\sum_{t=1}^{T} F_t$, i.e., $\mathcal{X}^* = \{(u,0) \mid -\frac{1}{2} \le u \le \frac{1}{2}\}$, which locates at the $x$-axis. For conciseness of analysis, we instantiate OMD with L2-regularization, which results in the simple OGD algorithm (McMahan, 2011). We start at an arbitrary point $\boldsymbol{x}_1 = (u_1, v_1) \in \mathcal{X}$ satisfying $v_1 > 0$. At each round $t$, suppose the decision $\boldsymbol{x}_t = (u_t, v_t)$, then the gradient of each objective w.r.t. $\boldsymbol{x}_t$ takes

$$\boldsymbol{g}_t^1 = \begin{cases} (2u_t + 4, & 2v_t + 2), & t = 2k - 1; \\ (2u_t, & 2v_t - 2), & t = 2k. \end{cases} \qquad \boldsymbol{g}_t^2 = \begin{cases} (2u_t, & 2v_t - 2), & t = 2k - 1; \\ (2u_t - 4, & 2v_t + 2), & t = 2k. \end{cases}$$

Since $0 \le v_t \le \frac{1}{2}$, we observe that the second entry of either gradient alternates between positive and negative. By using min-norm, the composite weights $\boldsymbol{\lambda}_t$ can be computed as

$$\boldsymbol{\lambda}_t = \begin{cases} ((1 - u_t - v_t)/4, & (3 + u_t + v_t)/4), & t = 2k - 1; \\ ((3 - u_t + v_t)/4, & (1 + u_t - v_t)/4), & t = 2k. \end{cases}$$

We observe that both entries of composite weights alternative between above $\frac{1}{2}$ and below $\frac{1}{2}$, and $\|\boldsymbol{\lambda}_{t+1} - \boldsymbol{\lambda}_t\|_1 \ge 1$. Recall that $\|\boldsymbol{\lambda}_t\|_1 = 1$, hence the composite weights at two consecutive rounds change radically. The resulting composite gradient takes

$$\boldsymbol{g}_t^{comp} = \begin{cases} (u_t - v_t + 1, & -u_t + v_t - 1), & t = 2k - 1; \\ (-u_t - v_t - 1, & -u_t - v_t - 1), & t = 2k. \end{cases}$$

The fluctuating composite weights mix with the positive and negative second entries of gradients, making the second entry of $\boldsymbol{g}_t^{comp}$ always negative, i.e., $-u_t + v_t - 1 < 0$ and $-u_t - v_t - 1 < 0$. Hence $\boldsymbol{g}_t^{comp}$ always drives $\boldsymbol{x}_t$ away from the Pareto set $\mathcal{X}^*$ that coincides with the $x$-axis. This essentially reversely optimizes the loss, hence increasing the regret. In fact, we can prove that it even incurs a linear regret. Due to the lack of space, we leave the proof of linear regret when $T$ is an odd number in Appendix H. The above results of the problem instance are summarized as follows.

**Proposition 2.** *For OMD equipped with vanilla min-norm, there exists a multi-objective online convex optimization problem, in which the resulting algorithm incurs a linear regret.*

***Remark.*** Stability is a basic requirement to ensure meaningful regrets in online learning (McMahan, 2017). In the single-objective setting, directly regularizing the iterate $\boldsymbol{x}_t$ (e.g., OMD) is enough. However, as shown in the above analysis, merely regularizing $\boldsymbol{x}_t$ is not enough to attain sublinear regrets in the multi-objective setting, since there is another source of instability, i.e., the composite weights, that affects the direction of composite gradients. Therefore, in multi-objective online learning, besides regularizing the iterates, we also need to explicitly regularize the composite weights.

## 4.2 THE ALGORITHM

Enlightened by the design of regularization in FTRL (McMahan, 2017), we consider the regularizer $r(\boldsymbol{\lambda}, \boldsymbol{\lambda}_0)$, where $\boldsymbol{\lambda}_0$ is the pre-defined composite weights that may reflect the user preference. This results in a new solver called *min-regularized-norm*, i.e.,

$$\boldsymbol{\lambda}_t = \underset{\boldsymbol{\lambda} \in \mathcal{S}_m}{\arg\min} \|\nabla F_t(\boldsymbol{x}_t)\boldsymbol{\lambda}\|_2^2 + \alpha_t\, r(\boldsymbol{\lambda}, \boldsymbol{\lambda}_0),$$

where $\alpha_t$ is the regularization strength. Equipping OMD with the new solver, we derive the proposed algorithm. Note that beyond the regularization on the iterate $\boldsymbol{x}_t$ that is intrinsic in online learning, there is another regularization on the composite weights $\boldsymbol{\lambda}_t$ in min-regularized-norm. Both regularizations are fundamental, and they together ensure stability in the multi-objective online setting. Hence we call the algorithm Doubly Regularized Online Mirror Multiple Descent (DR-OMMD).

In principle, $r$ can take various forms such as L1-norm, L2-norm, etc. Here we adopt L1-norm since it aligns well with the simplex constraint of $\lambda$. Min-regularized-norm can be computed very efficiently. When $m = 2$, it has a closed-form solution. Specifically, suppose the gradients at round $t$ are $\boldsymbol{g}_t^1$ and $\boldsymbol{g}_t^2$. Set $\gamma_L = (\boldsymbol{g}_2^\top(\boldsymbol{g}_2 - \boldsymbol{g}_1) - \alpha_t)/\|\boldsymbol{g}_2 - \boldsymbol{g}_1\|^2$ and $\gamma_R = (\boldsymbol{g}_2^\top(\boldsymbol{g}_2 - \boldsymbol{g}_1) + \alpha_t)/\|\boldsymbol{g}_2 - \boldsymbol{g}_1\|^2$. Given any $\boldsymbol{\lambda}_0 = (\gamma_0, 1 - \gamma_0) \in \mathcal{S}_2$, we can compute the composite weights $\boldsymbol{\lambda}_t$ as $(\gamma_t, 1 - \gamma_t)$ where

$$\gamma_t = \max\{\min\{\gamma_t'', 1\}, 0\}, \quad \text{where } \gamma_t'' = \max\{\min\{\gamma_0, \gamma_R\}, \gamma_L\}.$$

When $m > 2$, since the constraint $\mathcal{S}_m$ is a simplex, we can introduce a Frank-Wolfe solver (Jaggi, 2013) (see detailed protocol in Appendix E.1). We also discuss the L2-norm case in Appendix E.2.

Compared to vanilla min-norm, the composite weights in min-regularized-norm are not fully determined by the adversarial gradients. The resulting relative stability of composite weights makes the composite gradients more robust to the adversarial environment. In the following, we give a general analysis and prove that DR-OMMD indeed guarantees sublinear regrets.

### 4.3 Theoretical Analysis

Our analysis is based on two conventional assumptions (Jadbabaie et al., 2015; Hazan et al., 2016).

**Assumption 1.** The regularization function $R$ is 1-strongly convex. In addition, the Bregman divergence is $\gamma$-Lipschitz continuous, i.e., $B_R(\boldsymbol{x}, \boldsymbol{z}) - B_R(\boldsymbol{y}, \boldsymbol{z}) \leq \gamma \|\boldsymbol{x} - \boldsymbol{y}\|, \forall \boldsymbol{x}, \boldsymbol{y}, \boldsymbol{z} \in \mathrm{dom} R$, where $\mathrm{dom} R$ is the domain of $R$ and satisfies $\mathcal{X} \subset \mathrm{dom} R \subset \mathbb{R}^n$.

**Assumption 2.** There exists some finite $G > 0$ such that for each $i \in \{1, \ldots, m\}$, the $i$-th loss $f_t^i$ at each round $t \in \{1, \ldots, T\}$ is differentiable and $G$-Lipschitz continuous w.r.t. $\|\cdot\|_2$, i.e., $|f_t^i(\boldsymbol{x}) - f_t^i(\boldsymbol{x}')| \leq G\|\boldsymbol{x} - \boldsymbol{x}'\|_2$. Note that in the convex setting, this assumption leads to bounded gradients, i.e., $\|\nabla f_t^i(\boldsymbol{x})\|_2 \leq G$ for any $t \in \{1, \ldots, T\}, i \in \{1, \ldots, m\}, \boldsymbol{x} \in \mathcal{X}$.

**Theorem 1.** *Suppose the diameter of $\mathcal{X}$ is $D$. Assume $F_t$ is bounded, i.e., $|f_t^i(\boldsymbol{x})| \leq F, \forall \boldsymbol{x} \in \mathcal{X}, t \in \{1, \ldots, T\}, i \in \{1, \ldots, m\}$. For any $\boldsymbol{\lambda}_0 \in \mathcal{S}_m$, DR-OMMD attains*

$$R(T) \leq \frac{\gamma D}{\eta_T} + \sum_{t=1}^{T} \frac{\eta_t}{2} (\|\nabla F_t(\boldsymbol{x}_t)\boldsymbol{\lambda}_t\|_2^2 + \frac{4F}{\eta_t} \|\boldsymbol{\lambda}_t - \boldsymbol{\lambda}_0\|_1).$$

***Remark.*** When $\eta_t = \frac{\sqrt{2\gamma D}}{G\sqrt{T}}$ or $\frac{\sqrt{2\gamma D}}{G\sqrt{t}}$, $\alpha_t = \frac{4F}{\eta_t}$, the bound attains $O(\sqrt{T})$. It matches the optimal single-objective bound w.r.t. $T$ (Hazan et al., 2016) and is tight w.r.t. $m$ (justified in Appendix F.2).

**Comparison with linearization.** Linearization with fixed weights $\boldsymbol{\lambda}_0 \in \mathcal{S}_m$ essentially optimizes the scalar loss $\boldsymbol{\lambda}_0^\top F_t$ with gradient $\boldsymbol{g}_t = \nabla F_t(\boldsymbol{x}_t)\boldsymbol{\lambda}_0$. From OMD's tight bound (Theorem 6.8 in (Orabona, 2019)), we can derive a bound $\frac{\gamma D}{\eta_T} + \sum_{t=1}^{T} \frac{\eta_t}{2} \|\nabla F_t(\boldsymbol{x}_t)\boldsymbol{\lambda}_0\|_2^2$ for linearization. In comparison, when $\alpha_t = \frac{4F}{\eta_t}$, DR-OMMD attains a regret bound $\frac{\gamma D}{\eta_T} + \sum_{t=1}^{T} \frac{\eta_t}{2} \min_{\boldsymbol{\lambda} \in \mathcal{S}_m}\{\|\nabla F_t(\boldsymbol{x}_t)\boldsymbol{\lambda}\|_2^2 + \alpha_t \|\boldsymbol{\lambda} - \boldsymbol{\lambda}_0\|_1\}$, which is smaller than that of linearization. Note that although the bound of linearization refers to single-objective regret $R(T)$, the comparison is reasonable due to the consistency of the two regret metrics, i.e., $R_{\mathrm{II}}(T) = \max\{R(T), 0\}$ when $m = 1$, as proved in Proposition 1. In the following, we further investigate the margin in the two-objective setting with linear losses. Suppose the loss functions are $f_t^1(\boldsymbol{x}) = \boldsymbol{x}^\top \boldsymbol{g}_t^1$ and $f_t^2(\boldsymbol{x}) = \boldsymbol{x}^\top \boldsymbol{g}_t^2$ for some vectors $\boldsymbol{g}_t^1, \boldsymbol{g}_t^2 \in \mathbb{R}^n$ at each round. Then we can show that the margin is at least (see Appendix F.3 for the detailed proof)

$$M \geq \sum_{t=1}^{T} \frac{\eta_t}{4} \|\boldsymbol{\lambda}_t - \boldsymbol{\lambda}_0\|_2^2 \cdot \|\boldsymbol{g}_t^1 - \boldsymbol{g}_t^2\|_2^2,$$

which indicates the benefit of DR-OMMD. Specifically, while linearization requires adequate $\boldsymbol{\lambda}_0$, DR-OMMD selects more proper $\boldsymbol{\lambda}_t$ adaptively; the advantange is more obvious as the gradients of different objectives vary wildly. This matches our intuition that linearization suffers from conflict gradients (Yu et al., 2020), while DR-OMMD can alleviate the conflict by pursuing common descent.

## 5 Experiments

In this section, we conduct experiments to compare DR-OMMD with two baselines: (i) *linearization* performs single-objective online learning on scalar losses $\boldsymbol{\lambda}_0^\top F_t$ with pre-defined fixed $\boldsymbol{\lambda}_0 \in \mathcal{S}_m$; (ii) *min-norm* equips OMD with vanilla min-norm (Désidéri, 2012) for gradient composition.

### 5.1 Convex Experiments: Adaptive Regularization

Many real-world online scenarios adopt regularization to avoid overfitting. A standard scheme is to add a term $r(\boldsymbol{x})$ to the loss $f_t(\boldsymbol{x})$ at each round and optimize the regularized loss $f_t(\boldsymbol{x}) + \sigma r(\boldsymbol{x})$ (McMahan, 2011), where $\sigma$ is a pre-defined fixed hyperparameter. The formalism of multi-objective online learning provides a novel way of regularization. As $r(\boldsymbol{x})$ measures model complexity, it can

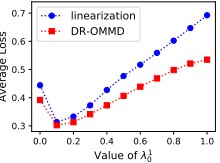 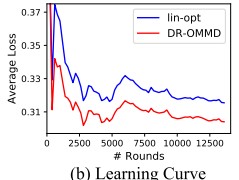 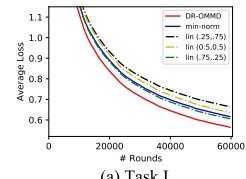 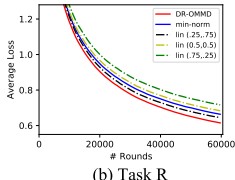

(a) Effect of Preference    (b) Learning Curve      (a) Task L    (b) Task R

Figure 1: Results to verify the effectiveness of adaptive regularization on *protein*. (a) Performance of DR-OMMD and linearization under varying $\boldsymbol{\lambda}_0 = (\lambda_0^1, 1 - \lambda_0^1)$. (b) Performance using the optimal weights $\boldsymbol{\lambda}_0 = (0.1, 0.9)$.

Figure 2: Results to verify the effectiveness of DR-OMMD in the non-convex setting. The two plots show the performance of DR-OMMD and various baselines on both tasks (Task L and Task R) of MultiMNIST.

be regarded as the second objective alongside the primary goal $f_t(\boldsymbol{x})$. We can augment the loss to $F_t(\boldsymbol{x}) = (f_t(\boldsymbol{x}), r(\boldsymbol{x}))$ and thereby cast regularized online learning into a two-objective problem. Compared to the standard scheme, our approach chooses $\sigma_t = \lambda_t^2 / \lambda_t^1$ in an adaptive way.

We use two large-scale online benchmark datasets. (i) *protein* is a bioinformatics dataset for protein type classification (Wang, 2002), which has 17 thousand instances with 357 features. (ii) *covtype* is a biological dataset collected from a non-stationary environment for forest cover type prediction (Blackard & Dean, 1999), which has 50 thousand instances with 54 features. We set the logistic classification loss as the first objective, and the squared L2-norm of model parameters as the second objective. Since the ultimate goal of regularization is to lift predictive performance, we measure the average loss, i.e., $\sum_{t \leq T} l_t(\boldsymbol{x}_t)/T$, where $l_t(\boldsymbol{x}_t)$ is the classification loss at round $t$.

We adopt a L2-norm ball centered at the origin with diameter $K = 100$ as the decision set. The learning rates are decided by a grid search over $\{0.1, 0.2, \ldots, 3.0\}$. For DR-OMMD, the parameter $\alpha_t$ is simply set as $0.1$. For fixed regularization, the strength $\sigma = (1 - \lambda_0^1)/\lambda_0^1$ is determined by some $\lambda_0^1 \in [0, 1]$, which is exactly *linearization* with weights $\boldsymbol{\lambda}_0 = (\lambda_0^1, 1 - \lambda_0^1)$. We run both algorithms with varying $\lambda_0^1 \in \{0, 0.1, ..., 1\}$. In Figure 1, we plot (a) their final performance w.r.t. the choice of $\boldsymbol{\lambda}_0$ and (b) their learning curves with desirable $\boldsymbol{\lambda}_0$ (e.g., $(0.1, 0.9)$ on *protein*). Other results are deferred to the appendix due to the lack of space. The results show that DR-OMMD consistently outperforms fixed regularization; the gap becomes more significant when $\boldsymbol{\lambda}_0$ is not properly set.

## 5.2 Non-Convex Experiments: Deep Multi-Task Learning

We use MultiMNIST (Sabour et al., 2017), which is a multi-task version of the MNIST dataset for image classification and commonly used in deep multi-task learning (Sener & Koltun, 2018; Lin et al., 2019). In MultiMNIST, each sample is composed of a random digit image from MNIST at the top-left and another image at the bottom-right. The goal is to classify the digit at the top-left (*task L*) and that at the bottom-right (*task R*) at the same time.

We follow (Sener & Koltun, 2018)'s setup with LeNet. Learning rates in all methods are selected via grid search over $\{0.0001, 0.001, 0.01, 0.1\}$. For linearization, we examine different weights $(0.25, 0.75)$, $(0.5, 0.5)$, and $(0.75, 0.25)$. For DR-OMMD, $\alpha_t$ is set according to Theorem 1, and the initial weights are simply set as $\boldsymbol{\lambda}_0 = (0.5, 0.5)$. Note that in the online setting, samples arrive in a sequential manner, which is different from offline experiments where sample batches are randomly sampled from the training set. Figure 2 compares the average cumulative loss of all the examined methods. We also measure two conventional metrics in offline experiments, i.e., the training loss and test loss (Reddi et al., 2018); the results are similar and deferred to the appendix due to the lack of space. The results show that DR-OMMD outperforms counterpart algorithms using min-norm or linearization in all metrics on both tasks, validating its effectiveness in the non-convex setting.

## 6 Conclusions

In this paper, we give a systematic study of multi-objective online learning, encompassing a novel framework, a new algorithm, and corresponding non-trivial theoretical analysis. We believe that this work paves the way for future research on more advanced multi-objective optimization algorithms, which may inspire the design of new optimizers for multi-task deep learning.

ACKNOWLEDGMENTS

This work was supported in part by the National Key Research and Development Program of China No. 2020AAA0106300 and National Natural Science Foundation of China No. 62250008. This work was also supported by Ant Group through Ant Research Intern Program. We would like to thank Wenliang Zhong, Jinjie Gu, Guannan Zhang and Jiaxin Liu for generous support on this project.

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

APPENDIX

The appendix is organized as follows. Appendix A reviews related work. Appendix B validates the correctness of our definition of PSG. Appendix C discusses the domain of the comparator in S-PSG, indicating that it makes no difference whether the comparator is selected from the Pareto optimal set or from the whole domain. Appendix D provides the detailed derivation of the equivalent form of $R_{\mathrm{II}}(T)$. Appendix E discusses how to efficiently compute the composition weights for the min-regularized-norm solver. Appendix F discusses the order of DR-OMMD's regret bound with fixed or adaptive learning rate, shows the tightness of the derived bound, and provides more details on the regret comparison between DR-OMMD and linearization. Appendix G supplements more details in the experimental setup and empirical results. Appendix H and I provide detailed proofs of the remaining theoretical claims in the main paper. Finally, Appendix J supplements regret analysis of DR-OMMD in the strongly convex setting.

## A    RELATED WORK

In this section, we review previous work in some related fields, i.e., online learning, multi-objective optimization, multi-objective multi-armed bandits, and multi-objective Bayesian optimization.

### A.1    ONLINE LEARNING

Online learning arms to make sequential predictions for streaming data. Please refer to the introduction books (Hazan et al., 2016; Orabona, 2019) for more background knowledges.

Most of the previous works on online learning are conducted in the single-objective setting. As far as we are concerned, there are only two lines of work concerning multi-objective learning. The first line of works provides a multi-objective perspective of the prediction-with-expert-advice (PEA) problem (Koolen, 2013; Koolen & Van Erven, 2015). Specifically, they view each individual expert as a multi-objective criterion, and characterize the Pareto optimal trade-offs among different experts. These works have two main distinctions from our proposed MO-OCO. First, they are still built upon the original PEA problem where the payoff of each expert (or decision) is a scalar, while we focus on vectoral payoffs. Second, their framework is restricted to an absolute loss game, whereas our framework is general and can be applied to any coordinate-wise convex loss functions.

The second line of work studies online learning with vectoral payoffs via Blackwell approachability (Blackwell, 1956; Mannor et al., 2014; Abernethy et al., 2011). In their framework, the learner is given a target set $\mathcal{T} \subset \mathbb{R}^m$ and its goal is to generate decisions $\{\boldsymbol{x}_t\}_{t=1}^T$ to minimize the distance between the average loss $\sum_{t=1}^T l_t(\boldsymbol{x}_t)/T$ and the target set $\mathcal{T}$. There are two major differences between Blackwell approachability and our proposed MO-OCO: previous works on Blackwell approachability are zero-order methods and the target set $\mathcal{T}$ is often known beforehand (also see the discussion in (Busa-Fekete et al., 2017)), while in MO-OCO we intend to develop a first-order method to reach the unknown Pareto front.

### A.2    MULTI-OBJECTIVE OPTIMIZATION

Multi-objective optimization aims to optimize multiple objectives concurrently. Most of the previous works on multi-objective optimization are conducted in the offline setting, including the batch optimization setting (Désidéri, 2012; Liu et al., 2021) and the stochastic optimization setting (Sener & Koltun, 2018; Lin et al., 2019; Yu et al., 2020; Chen et al., 2020; Javaloy & Valera, 2021). These methods are based on gradient composition, and have shown very promising results in multi-task learning applications.

Despite the existence of previous works on multi-objective optimization, as the first work of multi-objective optimization in the OCO setting, our work is largely different from them in three aspects. First, we contribute the first formal framework of multi-objective online convex optimization. In particular, our framework is based on a novel equivalent transformation of the PSG metric, which is intrinsically different from previous offline optimization frameworks. Second, we provide a showcase in which a commonly used method in the offline setting, namely min-norm (Désidéri, 2012; Sener & Koltun, 2018), fail to attain sublinear regret in online setting. Our proposed min-regularized-norm

is a novel design when tailoring offline methods to the online setting. Third, the regret analysis of multi-objective online learning is intrinsically different from the convergence analysis in the offline setting (Yu et al., 2020).

### A.3 Multi-Objective Multi-Armed Bandits

Another branch of related works study multi-objective optimization in the multi-armed bandits setting (Busa-Fekete et al., 2017; Tekin & Turğay, 2018; Turgay et al., 2018; Lu et al., 2019a; Degenne et al., 2019). Among these works, the most relevant one to ours is (Turgay et al., 2018), which introduces the Pareto suboptimality gap (PSG) metric to characterize the multi-objective regret in the bandits setting, and proposes a zero-order zooming algorithm to minimize the regret.

In this work, our regret definition also utilizes the PSG metric (Turgay et al., 2018). However, as the first study of multi-objective optimization in the OCO setting, our work is intrinsically different from these previous works in the following aspects. First, as PSG is a zero-order metric, we perform a novel equivalent transformation, making it amenable to the OCO setting. Second, our proposed algorithm is a first-order multiple gradient algorithm, whose design principles are completely distinct from zero-order algorithms. For example, the concept of the stability of composite weights does not even exist in the design of previous zero-order methods for multi-objective bandits (Turgay et al., 2018; Lu et al., 2019a). Third, the regret analysis of MO-OCO is intrinsically different from that in the bandits setting.

### A.4 Multi-Objective Bayesian Optimization

The final area related to our work is multi-objective Bayesian optimization (Zhang & Golovin, 2020; Konakovic Lukovic et al., 2020; Chowdhury & Gopalan, 2021; Maddox et al., 2021; Daulton et al., 2022), which studies Bayesian optimization with vector-valued feedback. There are two branches of works in this area, using different notions of regret. The first branch is based on scalarization, which adopts the expectation of the gap between scalarized losses over some given distribution (Chowdhury & Gopalan, 2021) as the regret. In this approach, the distribution of scalarization can be understood as a set of preference, which needs to be known beforehand. The second branch is based on Pareto optimality (Zhang & Golovin, 2020), which uses hypervolume as the discrepancy metric and adopt the gap between the true Pareto front and the estimated Pareto front as the regret.

As the first work on multi-objective optimization in the OCO setting, our work is largely different from these works in the following aspects. First, the regret definitions are different. Specifically, compared to the first branch based on scalarization, our regret definition is purely motivated by Pareto optimality, which does not need any preference in advance; compared to the second branch using hypervolume, we note that hypervolume is mainly used for Pareto front approximation, which is unsuitable to our adversarial setting where the goal is to impose the cumulative loss to reach the Pareto front. Second, multi-objective Bayesian optimization is conducted in a stochastic setting, which typically assumes that the losses follow some Gaussian distribution, whereas our work is conducted in the adversarial setting where the losses can be generated arbitrarily.

## B An Equivalent Definition of PSG

Recall that in Definition 3, we formulate the PSG metric as a constrained optimization problem. We note that, since the PSG metric is based on the notion of "non-dominance" (Turgay et al., 2018), its most direct form is actually

$$\Delta'(\boldsymbol{x}; \mathcal{K}^*, F) = \inf_{\epsilon \geq 0} \epsilon,$$

$$\text{s.t.} \quad \forall \boldsymbol{x}'' \in \mathcal{K}^*, \exists i \in \{1, \ldots, m\}, f^i(\boldsymbol{x}) - \epsilon < f^i(\boldsymbol{x}'')$$

$$\text{or} \quad \forall i \in \{1, \ldots, m\}, f^i(\boldsymbol{x}) - \epsilon = f^i(\boldsymbol{x}'').$$

At the first glance, the above definition seems to be quite different from Definition 3, since it has an extra condition "$\forall i \in \{1, \ldots, m\}, f^i(\boldsymbol{x}) - \epsilon = f^i(\boldsymbol{x}'')$". In the following, we prove that both definitions actually yield the same value due to the infimum operation on $\epsilon$.

Specifically, for any possible pair $(\boldsymbol{x}, \mathcal{K}^*, F)$, we denote $\Delta'(\boldsymbol{x}; \mathcal{K}^*, F) = \epsilon_0'$ and $\Delta(\boldsymbol{x}; \mathcal{K}^*, F) = \epsilon_0$. By comparing the constraints of both definitions, it is obvious that $\epsilon_0$ must satisfy the constraint

of $\Delta'(\boldsymbol{x}; \mathcal{K}^*, F)$, hence the infimum operation guarantees that $\epsilon'_0 \leq \epsilon_0$. It remains to prove that $\epsilon'_0 \geq \epsilon_0$. To this end, we only need to show that $\epsilon'_0 + \xi$ satisfies the constraint of $\Delta(\boldsymbol{x}; \mathcal{K}^*, F)$ for any $\xi > 0$. Consider an arbitrary $\boldsymbol{x}'' \in \mathcal{K}^*$. From the definition of $\Delta'(\boldsymbol{x}; \mathcal{K}^*, F)$, we know that either $\exists i \in \{1, \ldots, m\}, f^i(\boldsymbol{x}) - \epsilon'_0 < f^i(\boldsymbol{x}'')$ or $\forall i \in \{1, \ldots, m\}, f^i(\boldsymbol{x}) - \epsilon'_0 = f^i(\boldsymbol{x}'')$. Whichever condition holds, we must have $\exists i \in \{1, \ldots, m\}, f^i(\boldsymbol{x}) - \epsilon'_0 - \xi < f^i(\boldsymbol{x}'')$ for any $\xi > 0$. Since it holds for any $\boldsymbol{x}'' \in \mathcal{K}^*$, $\epsilon'_0 + \xi$ lies in the feasible region of $\Delta(\boldsymbol{x}; \mathcal{K}^*, F)$, hence we have $\epsilon_0 \leq \epsilon'_0 + \xi, \forall \xi > 0$ and thus $\epsilon_0 \leq \epsilon'_0$. In summary, we have $\Delta'(\boldsymbol{x}; \mathcal{K}^*, F) = \Delta(\boldsymbol{x}; \mathcal{K}^*, F)$ for any pair $(\boldsymbol{x}, \mathcal{K}^*, F)$.

## C  DISCUSSION ON THE DOMAIN OF THE COMPARATOR IN S-PSG

Recall that in Definition 4, the comparator $\boldsymbol{x}'$ in S-PSG is selected from the Pareto optimal set $\mathcal{X}^*$ of the cumulative loss $\sum_{t=1}^T F_t$. This actually stems from the original definition of PSG (Turgay et al., 2018), which uses the Pareto optimal set as the comparator set. In fact, comparing with Pareto optimal decisions in $\mathcal{X}^*$ is already enough to measure the suboptimality of any decision sequence $\{\boldsymbol{x}_t\}_{t=1}^T$. The reason is that, for any non-optimal decision $\boldsymbol{x}' \in \mathcal{X} - \mathcal{X}^*$, there must exist some Pareto optimal decision $\boldsymbol{x}'' \in \mathcal{X}^*$ that dominates $\boldsymbol{x}'$, hence the suboptimality metric does not need to compare with this non-optimal decision $\boldsymbol{x}'$. In other words, even if we extend the comparator set in S-PSG to the whole domain $\mathcal{X}$, the modified form will be equivalent to the original form based on the Pareto optimal set $\mathcal{X}^*$. In the following, we strictly prove this equivalence $\Delta(\{\boldsymbol{x}_t\}_{t=1}^T; \mathcal{X}, \{F_t\}_{t=1}^T) = \Delta(\{\boldsymbol{x}_t\}_{t=1}^T; \mathcal{X}^*, \{F_t\}_{t=1}^T)$.

Specifically, we modify the definition of S-PSG and let the comparator domain $\mathcal{X}'$ be any subset of the decision domain $\mathcal{X}$, i.e.,

$$\Delta(\{\boldsymbol{x}_t\}_{t=1}^T; \mathcal{X}', \{F_t\}_{t=1}^T) = \inf_{\epsilon \geq 0} \epsilon, \text{ s.t. } \forall \boldsymbol{x}'' \in \mathcal{X}', \exists i \in \{1, \ldots, m\}, \sum_{t=1}^T f_t^i(\boldsymbol{x}_t) - \epsilon < \sum_{t=1}^T f_t^i(\boldsymbol{x}'').$$

Then the modified regret based on the whole domain $\mathcal{X}$ takes $R'_{\mathrm{II}}(T) = \Delta(\{\boldsymbol{x}_t\}_{t=1}^T; \mathcal{X}, \{F_t\}_{t=1}^T)$. Now we begin to prove the equivalence $\Delta(\{\boldsymbol{x}_t\}_{t=1}^T; \mathcal{X}, \{F_t\}_{t=1}^T) = \Delta(\{\boldsymbol{x}_t\}_{t=1}^T; \mathcal{X}^*, \{F_t\}_{t=1}^T)$. For any $\mathcal{X}' \subset \mathcal{X}$, let $\mathcal{E}(\mathcal{X}')$ denote the constraint of $\Delta(\{\boldsymbol{x}_t\}_{t=1}^T; \mathcal{X}', \{F_t\}_{t=1}^T)$, i.e.,

$$\mathcal{E}(\mathcal{X}') = \{\epsilon \geq 0 \mid \forall \boldsymbol{x}'' \in \mathcal{X}', \exists i \in \{1, \ldots, m\}, \sum_{t=1}^T f_t^i(\boldsymbol{x}_t) - \epsilon < \sum_{t=1}^T f_t^i(\boldsymbol{x}'')\},$$

then $\Delta(\{\boldsymbol{x}_t\}_{t=1}^T; \mathcal{X}', \{F_t\}_{t=1}^T) = \inf \mathcal{E}(\mathcal{X}')$. Hence, we just need to prove $\inf \mathcal{E}(\mathcal{X}) = \inf \mathcal{E}(\mathcal{X}^*)$.

On the one hand, since $\mathcal{X}^* \subset \mathcal{X}$, from the above definition of S-PSG, it is easy to check that for any $\epsilon \in \mathcal{E}(\mathcal{X})$, it must satisfy $\epsilon \in \mathcal{E}(\mathcal{X}^*)$. Hence, we have $\mathcal{E}(\mathcal{X}) \subset \mathcal{E}(\mathcal{X}^*)$.

On the other hand, given any $\epsilon \in \mathcal{E}(\mathcal{X}^*)$, we now check that $\epsilon \in \mathcal{E}(\mathcal{X})$. To this end, we consider an arbitrary point $\boldsymbol{x}'' \in \mathcal{X}$ in two cases. (i) If $\boldsymbol{x}'' \in \mathcal{X}^*$, since $\epsilon \in \mathcal{E}(\mathcal{X}^*)$, we naturally have $\sum_{t=1}^T f_t^{i_0}(\boldsymbol{x}_t) - \epsilon < \sum_{t=1}^T f_t^{i_0}(\boldsymbol{x}'')$ for some $i_0$. (ii) If $\boldsymbol{x}'' \notin \mathcal{X}^*$, since $\mathcal{X}^*$ is the Pareto optimal set of $\sum_{t=1}^T F_t$, there must exist some Pareto optimal decision $\hat{\boldsymbol{x}} \in \mathcal{X}^*$ that dominates $\boldsymbol{x}''$ w.r.t. $\sum_{t=1}^T F_t$, which means that $\sum_{t=1}^T f_t^i(\hat{\boldsymbol{x}}) \leq \sum_{t=1}^T f_t^i(\boldsymbol{x}'')$ for all $i \in \{1, ..., m\}$. Notice that $\epsilon \in \mathcal{E}(\mathcal{X}^*)$ gives $\sum_{t=1}^T f_t^{i_0}(\boldsymbol{x}_t) - \epsilon < \sum_{t=1}^T f_t^{i_0}(\hat{\boldsymbol{x}})$ for some $i_0$, hence in this case we also have $\sum_{t=1}^T f_t^{i_0}(\boldsymbol{x}_t) - \epsilon < \sum_{t=1}^T f_t^{i_0}(\boldsymbol{x}'')$. Combining the above two cases, we prove that $\epsilon \in \mathcal{E}(\mathcal{X})$, and consequently $\mathcal{E}(\mathcal{X}^*) \subset \mathcal{E}(\mathcal{X})$.

In summary, we have $\mathcal{E}(\mathcal{X}) = \mathcal{E}(\mathcal{X}^*)$, hence $\Delta(\{\boldsymbol{x}_t\}_{t=1}^T; \mathcal{X}, \{F_t\}_{t=1}^T) = \inf \mathcal{E}(\mathcal{X}) = \inf \mathcal{E}(\mathcal{X}^*) = \Delta(\{\boldsymbol{x}_t\}_{t=1}^T; \mathcal{X}^*, \{F_t\}_{t=1}^T)$. Therefore, it makes no difference whether the comparator in $R_{\mathrm{II}}(T)$ is generated from the Pareto optimal set $\mathcal{X}^*$ or from the whole domain $\mathcal{X}$.

## D  DERIVATION OF THE EQUIVALENT MULTI-OBJECTIVE REGRET FORM

In this section, We strictly derive the equivalent form of $R_{\mathrm{II}}(T)$ in Proposition 1, which is highly non-trivial and forms the basis of the subsequent algorithm design and theoretical analysis.

*Proof of Proposition 1.* Recall that the PSG metric used in $R_{\mathrm{II}}(T)$ is an extension of vanilla PSG to leverage any decision sequence. To motivate the analysis, we first investigate vanilla PSG $\Delta(\boldsymbol{x}; \mathcal{X}^*, F)$ that deals with a single decision $\boldsymbol{x}$, and derive a useful lemma as follows.

**Lemma 1.** *Vanilla PSG has an equivalent form, i.e.,*

$$\Delta(\boldsymbol{x}; \mathcal{X}^*, F) = \sup_{\boldsymbol{x}^* \in \mathcal{X}^*} \inf_{\boldsymbol{\lambda} \in \mathcal{S}_m} \boldsymbol{\lambda}^\top (F(\boldsymbol{x}) - F(\boldsymbol{x}))_+,$$

*where for any vector $\boldsymbol{l} = (l^1, ..., l^m) \in \mathbb{R}^m$, the truncation $(\boldsymbol{l})_+$ produces a vector whose $i$-th entry equals to $\max\{l^i, 0\}$ for all $i \in \{1, ..., m\}$.*

*Proof.* In the definition of PSG, the evaluated decision $\boldsymbol{x}$ is compared to all Pareto optimal points $\boldsymbol{x}' \in \mathcal{X}^*$. For any fixed comparator $\boldsymbol{x}' \in \mathcal{X}^*$, we define the **pair-wise suboptimality gap** w.r.t. $F$ between decisions $\boldsymbol{x}$ and $\boldsymbol{x}'$ as follows

$$\delta(\boldsymbol{x}; \boldsymbol{x}', F) = \inf_{\epsilon \geq 0}\{\epsilon \mid F(\boldsymbol{x}) - \epsilon \mathbf{1} \nsucc F(\boldsymbol{x}')\}.$$

Hence, PSG can be expressed as

$$\Delta(\boldsymbol{x}; \mathcal{X}^*, F) = \sup_{\boldsymbol{x}' \in \mathcal{X}^*} \delta(\boldsymbol{x}; \boldsymbol{x}', F).$$

To proceed, we analyze the pair-wise gap $\delta(\boldsymbol{x}; \boldsymbol{x}', F)$. From its definition, we know that $\delta(\boldsymbol{x}; \boldsymbol{x}', F)$ measures the minimal non-negative value that needs to be subtracted from each entry of $F(\boldsymbol{x})$ until it is not dominated by $\boldsymbol{x}'$. Now we consider two cases.

(i) If $F(\boldsymbol{x}) \nsucc F(\boldsymbol{x}')$, i.e., $f^{k_0}(\boldsymbol{x}) \leq f^{k_0}(\boldsymbol{x}')$ for some $k_0 \in \{1, ..., m\}$, nothing needs to be subtracted from $F(\boldsymbol{x})$ and we directly have $\delta(\boldsymbol{x}; \boldsymbol{x}', F) = 0$.

(ii) If $F(\boldsymbol{x}) \succ F(\boldsymbol{x}')$, we have $f^k(\boldsymbol{x}) \geq f^k(\boldsymbol{x}')$ for all $k \in \{1, ..., m\}$, which obviously violates the condition $F(\boldsymbol{x}) - \epsilon \mathbf{1} \nsucc F(\boldsymbol{x}')$ when $\epsilon = 0$. Now let us gradually increase $\epsilon$ from zero. Notice that such a condition holds only when there there exists some $k_0$ satisfying $f^{k_0}(\boldsymbol{x}) - \epsilon \leq f^{k_0}(\boldsymbol{x}')$, or equivalently $\epsilon \geq f^{k_0}(\boldsymbol{x}) - f^{k_0}(\boldsymbol{x}')$. Hence, in this case, we have $\delta(\boldsymbol{x}; \boldsymbol{x}', F) = \min_{k \in \{1, ..., m\}}\{f^k(\boldsymbol{x}) - f^k(\boldsymbol{x}')\}$.

Combining the above two cases, we derive an equivalent form of the pair-wise suboptimality gap. Specifically, we can easily check that the following form holds for both cases, i.e.,

$$\delta(\boldsymbol{x}; \boldsymbol{x}', F) = \min_{k \in \{1, ..., m\}} \max\{f^k(\boldsymbol{x}) - f^k(\boldsymbol{x}'), 0\}.$$

To relate the above form with $F$, denote $\mathcal{U}_m = \{\boldsymbol{e}_k \mid 1 \leq k \leq m\}$ as the set of all unit vector in $\mathbb{R}^m$, then we equivalently have

$$\delta(\boldsymbol{x}; \boldsymbol{x}', F) = \min_{\boldsymbol{\lambda} \in \mathcal{U}_m} \boldsymbol{\lambda}^\top (F(\boldsymbol{x}) - F(\boldsymbol{x}'))_+.$$

Now the calculation of $\delta(\boldsymbol{x}; \boldsymbol{x}', F)$ is transformed into a minimization problem over $\boldsymbol{\lambda} \in \mathcal{U}_m$. Since $\mathcal{U}_m$ is a discrete set, we can apply a linear relaxation trick. Specifically, we now turn to minimize the scalar $p(\boldsymbol{\lambda}) = \boldsymbol{\lambda}^\top \max\{F(\boldsymbol{x}) - F(\boldsymbol{x}'), 0\}$ over the convex curvature of $\mathcal{U}_m$, which is exactly the probability simplex $\mathcal{S}_m = \{\boldsymbol{\lambda} \in \mathbb{R}^m \mid \boldsymbol{\lambda} \succeq \boldsymbol{0}, \|\boldsymbol{\lambda}\|_1 = 1\}$. Note that $\mathcal{U}_m$ contains all the vertexes of $\mathcal{S}_m$. Since $\inf_{\boldsymbol{\lambda} \in \mathcal{S}_m} p(\boldsymbol{\lambda})$ is a linear optimization problem, the minimal point $\boldsymbol{\lambda}^*$ must be a vertex of the simplex, i.e., $\boldsymbol{\lambda}^* \in \mathcal{U}_m$. Hence, the relaxed problem is equivalent to the original problem, namely,

$$\delta(\boldsymbol{x}; \boldsymbol{x}', F) = \min_{\boldsymbol{\lambda} \in \mathcal{U}_m} \boldsymbol{\lambda}^\top (F(\boldsymbol{x}) - F(\boldsymbol{x}'))_+ = \inf_{\boldsymbol{\lambda} \in \mathcal{S}_m} \boldsymbol{\lambda}^\top (F(\boldsymbol{x}) - F(\boldsymbol{x}'))_+.$$

Taking the supremum of both sides over $\boldsymbol{x}' \in \mathcal{X}^*$, we prove the lemma. ∎

The above lemma can be naturally extended to the sequence-wise variant S-PSG. Specifically, we can extend the pair-wise suboptimality gap $\delta(\boldsymbol{x}; \boldsymbol{x}', F)$ to measure any decision sequence, which now becomes

$$\delta(\{\boldsymbol{x}_t\}_{t=1}^T; \boldsymbol{x}', \{F_t\}_{t=1}^T) = \inf_{\epsilon \geq 0}\{\epsilon \mid \sum_{t=1}^T F_t(\boldsymbol{x}_t) - \epsilon \mathbf{1} \nsucc \sum_{t=1}^T F_t(\boldsymbol{x}')\}.$$

Then S-PSG can be expressed as

$$\Delta(\{\boldsymbol{x}_t\}_{t=1}^T; \mathcal{X}^*, \{F_t\}_{t=1}^T) = \sup_{\boldsymbol{x}^* \in \mathcal{X}^*} \delta(\{\boldsymbol{x}_t\}_{t=1}^T; \boldsymbol{x}^*, \{F_t\}_{t=1}^T).$$

Similar to the derivation of the above lemma, by investigating the relation between $\sum_{t=1}^T F_t(x_t)$ and $\sum_{t=1}^T F_t(x')$, we can derive an equivalent form of $\delta(\{\boldsymbol{x}_t\}_{t=1}^T; \boldsymbol{x}', \{F_t\}_{t=1}^T)$ as

$$\delta(\{\boldsymbol{x}_t\}_{t=1}^T; \boldsymbol{x}', \{F_t\}_{t=1}^T) = \min_{k \in \{1,...,m\}} \max\{\sum_{t=1}^T f_t^k(\boldsymbol{x}) - \sum_{t=1}^T f_t^k(\boldsymbol{x}'), 0\},$$

and further

$$\delta(\{\boldsymbol{x}_t\}_{t=1}^T; \boldsymbol{x}', \{F_t\}_{t=1}^T) = \inf_{\boldsymbol{\lambda} \in \mathcal{S}_m} \boldsymbol{\lambda}^\top (\sum_{t=1}^T F_t(\boldsymbol{x}_t) - \sum_{t=1}^T F_t(\boldsymbol{x}'))_+.$$

Hence, the S-PSG-based regret form can be expressed as

$$R_{\mathrm{II}}(T) = \sup_{\boldsymbol{x}^* \in \mathcal{X}^*} \inf_{\boldsymbol{\lambda} \in \mathcal{S}_m} \boldsymbol{\lambda}^\top (\sum_{t=1}^T F_t(\boldsymbol{x}_t) - \sum_{t=1}^T F_t(\boldsymbol{x}^*))_+.$$

The max-min form of $R_{\mathrm{II}}(T)$ has a truncation operation $(\cdot)_+$, which brings irregularity to the regret form. To handle the truncation operation, we utilize the following lemma:

**Lemma 2.** *(a) For any $\boldsymbol{l} \in \mathbb{R}^m$, we have $\inf_{\boldsymbol{\lambda} \in \mathcal{S}_m} \boldsymbol{\lambda}^\top (\boldsymbol{l})_+ = \max\{\inf_{\boldsymbol{\lambda} \in \mathcal{S}_m} \boldsymbol{\lambda}^\top \boldsymbol{l}, 0\}$.*
*(b) For any $h : \mathcal{X} \to \mathbb{R}$, we have $\sup_{\boldsymbol{x} \in \mathcal{X}} \max\{h(\boldsymbol{x}), 0\} = \max\{\sup_{\boldsymbol{x} \in \mathcal{X}} h(\boldsymbol{x}), 0\}$.*

*Proof.* To prove the first statement, we consider the following two cases.
(i) If $\boldsymbol{l} \succ \boldsymbol{0}$, then $(\boldsymbol{l})_+ = l$. For any $\boldsymbol{\lambda} \in \mathcal{S}_m$, we have $\boldsymbol{\lambda}^\top (\boldsymbol{l})_+ = \boldsymbol{\lambda}^\top \boldsymbol{l} > 0$. Taking the infimum over $\boldsymbol{\lambda} \in \mathcal{S}_m$ on both sides, we have $\inf_{\boldsymbol{\lambda}^\top \mathcal{S}_m} \boldsymbol{\lambda}^\top (\boldsymbol{l})_+ = \inf_{\boldsymbol{\lambda} \in \mathcal{S}_m} \boldsymbol{\lambda}^\top \boldsymbol{l} \geq 0$. Moreover, from the last equation we have $\max\{\inf_{\boldsymbol{\lambda} \in \mathcal{S}_m} \boldsymbol{\lambda}^\top \boldsymbol{l}, 0\} = \inf_{\boldsymbol{\lambda} \in \mathcal{S}_m} \boldsymbol{\lambda}^\top \boldsymbol{l}$, which proves the statement in this case.
(ii) If $\boldsymbol{l} \not\succ \boldsymbol{0}$, then $l^i \leq 0$ for some $i \in \{1, ..., m\}$. Set $\boldsymbol{e}_i$ as the $i$-th unit vector in $\mathbb{R}^m$, then we have $\boldsymbol{e}_i^\top \boldsymbol{l} \leq 0$. One the one hand, since $\boldsymbol{e}_i \in \mathcal{S}_m$, we have $\inf_{\boldsymbol{\lambda} \in \mathcal{S}_m} \boldsymbol{\lambda}^\top \boldsymbol{l} \leq \boldsymbol{e}_i^\top \boldsymbol{l} \leq 0$, and further $\max\{\inf_{\boldsymbol{\lambda} \in \mathcal{S}_m} \boldsymbol{\lambda}^\top \boldsymbol{l}, 0\} = 0$. On the other hand, notice that $\boldsymbol{e}_i^\top (\boldsymbol{l})_+ = 0$ and $\boldsymbol{\lambda}^\top (\boldsymbol{l})_+ \geq 0$ for any $\boldsymbol{\lambda} \in \mathcal{S}_m$, then $\inf_{\boldsymbol{\lambda} \in \mathcal{S}_m} \boldsymbol{\lambda}^\top (\boldsymbol{l})_+ = \boldsymbol{e}_i^\top (\boldsymbol{l})_+ = 0$. Hence, the statement also holds in this case.

To prove the second statement, we also consider two cases.
(i) If $h(\boldsymbol{x}_0) > 0$ for some $\boldsymbol{x}_0 \in \mathcal{X}$, then $\sup_{\boldsymbol{x} \in \mathcal{X}} h(\boldsymbol{x}) \geq h(\boldsymbol{x}_0) > 0$, and $\max\{\sup_{\boldsymbol{x} \in \mathcal{X}} h(\boldsymbol{x}), 0\} = \sup_{\boldsymbol{x} \in \mathcal{X}} h(\boldsymbol{x})$. Since we also have $\sup_{\boldsymbol{x} \in \mathcal{X}} \max\{h(\boldsymbol{x}), 0\} = \sup_{\boldsymbol{x} \in \mathcal{X}} h(\boldsymbol{x})$, the statement holds in this case.
(ii) If $h(\boldsymbol{x}) \leq 0$ for all $\boldsymbol{x} \in \mathcal{X}$, then $\sup_{\boldsymbol{x} \in \mathcal{X}} h(\boldsymbol{x}) \leq 0$, and thus $\max\{\sup_{\boldsymbol{x} \in \mathcal{X}} h(\boldsymbol{x}), 0\} = 0$. Meanwhile, for any $\boldsymbol{x} \in \mathcal{X}$, we have $\max\{h(\boldsymbol{x})\} = 0$, which validates the statement in this case. ∎

From the above lemma, we directly have

$$R_{\mathrm{II}}(T) = \sup_{\boldsymbol{x}^* \in \mathcal{X}^*} \max\{\inf_{\boldsymbol{\lambda} \in \mathcal{S}_m} \boldsymbol{\lambda}^\top (\sum_{t=1}^T F_t(\boldsymbol{x}_t) - \sum_{t=1}^T F_t(\boldsymbol{x}^*)), 0\}$$

$$= \max\{\sup_{\boldsymbol{x}^* \in \mathcal{X}^*} \inf_{\boldsymbol{\lambda} \in \mathcal{S}_m} \boldsymbol{\lambda}^\top (\sum_{t=1}^T F_t(\boldsymbol{x}_t) - \sum_{t=1}^T F_t(\boldsymbol{x}^*)), 0\},$$

which derives the desired equivalent form. ∎

# E  CALCULATION OF MIN-REGULARIZED-NORM

In this section, we discuss how to efficiently calculate the solutions to min-regularized-norm with L1-norm and L2-norm.

---

**Algorithm 2** Frank-Wolfe Solver for Min-Regularized-Norm with L1-Norm

1: **Initialize:** $\boldsymbol{\lambda}_t = (\gamma_t^1, \ldots, \gamma_t^m) = (\frac{1}{m}, \ldots, \frac{1}{m})$.
2: Compute the matrix $\mathbf{U} = \nabla F_t(\boldsymbol{x}_t)^\top \nabla F_t(\boldsymbol{x}_t)$, i.e., $\mathbf{U}^{ij} = \nabla f_t^i(\boldsymbol{x}_t)^\top \nabla f_t^j(\boldsymbol{x}_t), \forall i, j \in \{1, \ldots, m\}$.
3: **repeat**
4:     Select an index $k \in \arg\max_{i \in \{1, \ldots, m\}} \{\sum_{j=1}^m \gamma_t^j \mathbf{U}^{ij} + \alpha \operatorname{sgn}(\gamma_t^i - \gamma_0^i)\}$.
5:     Compute $\delta \in \arg\min_{0 \le \delta \le 1} \left\| \delta \nabla f_t^k(\boldsymbol{x}_t) + (1 - \delta) \nabla F_t(\boldsymbol{x}_t) \boldsymbol{\lambda}_t \right\|_2^2 + \alpha \| \delta(\boldsymbol{e}_k - \boldsymbol{\lambda}_t) + \boldsymbol{\lambda}_t - \boldsymbol{\lambda}_0 \|_1$.
6:     Update $\boldsymbol{\lambda}_t = (1 - \delta) \boldsymbol{\lambda}_t + \delta \boldsymbol{e}_k$.
7: **until** $\delta \sim 0$ **or** Number of Iteration Limits
8: **return** $\boldsymbol{\lambda}_t$.

---

### E.1 L1-Norm

Similar to (Sener & Koltun, 2018), we first consider the setting of two objectives, namely $m = 2$. In this case, for any $\boldsymbol{\lambda} = (\gamma, 1 - \gamma), \boldsymbol{\lambda}_0 = (\gamma_0, 1 - \gamma_0) \in \mathcal{S}_2$, the L1-regularization $\|\boldsymbol{\lambda} - \boldsymbol{\lambda}_0\|_1$ equals to $2|\gamma - \gamma_0|$. Hence min-regularized-norm with L1-norm at round $t$ reduces to $\boldsymbol{\lambda}_t = (\gamma_t, 1 - \gamma_t)$ where

$$\gamma_t \in \arg\min_{0 \le \gamma \le 1} \|\gamma \boldsymbol{g}_1 + (1 - \gamma) \boldsymbol{g}_2\|_2^2 + 2\alpha |\gamma - \gamma_0|.$$

Interestingly, the above problem has a closed-form solution.

**Proposition 3.** *Set* $\gamma_L = (\boldsymbol{g}_2^\top (\boldsymbol{g}_2 - \boldsymbol{g}_1) - \alpha) / \|\boldsymbol{g}_2 - \boldsymbol{g}_1\|_2^2$, *and* $\gamma_R = (\boldsymbol{g}_2^\top (\boldsymbol{g}_2 - \boldsymbol{g}_1) + \alpha) / \|\boldsymbol{g}_2 - \boldsymbol{g}_1\|_2^2$. *Then min-regularized-norm with L1-norm produces weights* $\boldsymbol{\lambda}_t = (\gamma_t, 1 - \gamma_t)$ *where*

$$\gamma_t = \max\{\min\{\gamma_t'', 1\}, 0\}, \quad \text{where } \gamma_t'' = \max\{\min\{\gamma_0, \gamma_R\}, \gamma_L\}.$$

*Proof.* We solve the following two quadratic sub-problems, i.e.,

$$\min_{0 \le \gamma \le \gamma_0} h_1(\gamma) = \|\gamma \boldsymbol{g}_1 + (1 - \gamma) \boldsymbol{g}_2\|_2^2 + 2\alpha(\gamma_0 - \gamma),$$

as well as

$$\min_{\gamma_0 \le \gamma \le 1} h_2(\gamma) = \|\gamma \boldsymbol{g}_1 + (1 - \gamma) \boldsymbol{g}_2\|_2^2 + 2\alpha(\gamma - \gamma_0).$$

It can be checked that in the former sub-problem, $h_1$ monotonously decreases on $(-\infty, \gamma_R]$ and increases on $[\gamma_R, +\infty)$; in the latter sub-problem, $h_2$ monotonously decreases on $(-\infty, \gamma_L]$ and increases on $[\gamma_L, +\infty)$. Since each sub-problem has its constraint ($[0, \gamma_0]$ or $[\gamma_0, 1]$), the solution to the original optimization problem can then be derived by comparing the optimal values of the two sub-problems with their constraints. Specifically, notice that $\gamma_L \le \gamma_R$ and $0 \le \gamma_0 \le 1$, and we can consider the following three cases.

(i) When $0 \le \gamma_0 \le \gamma_L \le \gamma_R$, then $h_1$ monotonously decreases on $[0, \gamma_0]$ and its minimum on $[0, \gamma_0]$ is $h_1(\gamma_0)$. Notice that $h_1(\gamma_0) = h_2(\gamma_0)$. For the sub-problem of $h_2$, we further consider two situations:
(i-a) If $\gamma_L \le 1$, then $\gamma_L \in [\gamma_0, 1]$, hence the minimum of $h_2$ on $[\gamma_0, 1]$ is $h_2(\gamma_L)$. Since $h_2(\gamma_L) \le h_2(\gamma_0) = h_1(\gamma_0)$, the minimal point of the original problem is $\gamma_L$, and hence $\gamma_t = \gamma_L$.
(i-b) If $\gamma_L > 1$, then $h_2$ monotonously decreases on $[\gamma_0, 1]$, and we surely have $h_2(1) \le h_2(\gamma_0) = h_1(\gamma_0)$. Hence $\gamma_t = 1$ in this situation.
Combining the above two situations, we have $\gamma_t = \min\{\gamma_L, 1\}$ in this case.

(ii) When $\gamma_L \le \gamma_R \le \gamma_0 \le 1$, then $h_2$ monotonously increases on $[\gamma_0, 1]$ and its minimum on $[\gamma_0, 1]$ is $h_2(\gamma_0)$. Notice that $h_1(\gamma_0) = h_2(\gamma_0)$. For the sub-problem of $h_1$, similar to the first case, we also consider two situations:
(ii-a) If $\gamma_R \ge 0$, then $\gamma_R \in [0, \gamma_0]$, hence the minimum of $h_1$ on $[0, \gamma_0]$ is $h_1(\gamma_R)$. Since $h_1(\gamma_R) \le h_1(\gamma_0) = h_2(\gamma_0)$, the minimal point of the original problem is $\gamma_R$, and hence $\gamma_t = \gamma_R$.
(ii-b) If $\gamma_R < 0$, then $h_1$ monotonously increases on $[0, \gamma_0]$. Hence we have $h_1(0) \le h_1(\gamma_0) = h_2(\gamma_0)$. Hence the solution to the original problem $\gamma_t = 0$.
Combining the above two situations, we have $\gamma_t = \max\{\gamma_R, 0\}$ in this case.

---

**Algorithm 3** Frank-Wolfe Solver for Min-Regularized-Norm with L2-Norm

---

1: **Initialize:** $\boldsymbol{\lambda}_t = (\gamma_t^1, \ldots, \gamma_t^m) = (\frac{1}{m}, \ldots, \frac{1}{m})$.
2: Compute the matrix $\mathbf{U} = \nabla F_t(\boldsymbol{x}_t)^\top \nabla F_t(\boldsymbol{x}_t)$, i.e., $\mathbf{U}^{ij} = \nabla f_t^i(\boldsymbol{x}_t)^\top \nabla f_t^j(\boldsymbol{x}_t), \forall i, j \in \{1, \ldots, m\}$.
3: **repeat**
4:     Select an index $k \in \arg\max_{i \in \{1,\ldots,m\}} \{\sum_{j=1}^m \gamma_t^j \mathbf{U}^{ij} + \alpha(\gamma_t^i - \gamma_0^i)\}$.
5:     Compute $\delta \in \arg\min_{0 < \delta \le 1} \|\delta \nabla f_t^k(\boldsymbol{x}_t)) + (1-\delta)\nabla F_t(\boldsymbol{x}_t)\boldsymbol{\lambda}_t\|_2^2 + \alpha\|\delta(\boldsymbol{e}_k - \boldsymbol{\lambda}_t) + \boldsymbol{\lambda}_t - \boldsymbol{\lambda}_0\|_2^2$, which has an analytical form

$$\delta = \max\{\min\{\frac{(\nabla F_t(\boldsymbol{x}_t)\boldsymbol{\lambda}_t - \nabla f_t^k(\boldsymbol{x}_t))^\top \nabla F_t(\boldsymbol{x}_t)\boldsymbol{\lambda}_t + \alpha\|\boldsymbol{e}_k - \boldsymbol{\lambda}_t\|_2^2}{\|\nabla F_t(\boldsymbol{x}_t)\boldsymbol{\lambda}_t - \nabla f_t^k(\boldsymbol{x}_t)\|_2^2 + \alpha(\boldsymbol{e}_k - \boldsymbol{\lambda}_t)^\top(\boldsymbol{\lambda}_t - \boldsymbol{\lambda}_0)}, 1\}, 0\}.$$

6:     Update $\boldsymbol{\lambda}_t = (1-\delta)\boldsymbol{\lambda}_t + \delta\boldsymbol{e}_k$.
7: **until** $\delta \sim 0$ **or** Number of Iteration Limits
8: **return** $\boldsymbol{\lambda}_t$.

---

(iii) When $\gamma_L < \gamma_0 < \gamma_R$, then $h_1$ monotonously decreases on $[0, \gamma_0]$ and $h_2$ monotonously increases on $[\gamma_0, 1]$. Hence each sub-problem attains its minimum at $\gamma_0$, and thus $\gamma_t = \gamma_0$.

Summarizing the above three cases gives

$$\gamma_t = \begin{cases} \min\{\gamma_L, 1\}, & \gamma_0 \le \gamma_L; \\ \max\{\gamma_R, 0\}, & \gamma_0 \ge \gamma_R; \\ \gamma_0, & \text{otherwise.} \end{cases}$$

We can further rewrite the above formula into a compact form as follows, which can be checked case-by-case.

$$\gamma_t = \max\{\min\{\gamma_t'', 1\}, 0\}, \quad \text{where } \gamma_t'' = \max\{\min\{\gamma_0, \gamma_R\}, \gamma_L\},$$

This gives the closed-form solution of min-regularized-norm when $m = 2$. ∎

Now that we have derived the closed-form solution to the min-regularized-norm solver with any two gradients, in principle, we can apply (Sener & Koltun, 2018)'s technique to efficiently compute the solution to the solver with more than two gradients. We provide the full procedure in Algorithm 2, which is an extension of (Sener & Koltun, 2018). By following the exact line search technique (Jaggi, 2013) in MGDA, we get our line search oracle as line 5 in Algorithm 2. The first term is the same as that in MGDA, and the second term is an extra L1-regularization term related to the design in Algorithm 1. Unlike the oracle of MGDA that has a closed-form solution by a reduction to the case of two gradients, the extra L1-norm term makes our oracle difficult to get a closed-form solution. The reason is that, such an extra term is the L1-norm of a $m$-dimension vector, hence it can not simply reduce to the case of two gradients. To proceed, we can directly apply numerical methods to get the solution (e.g. similar to the implementation in (Liu et al., 2021)).

### E.2 L2-NORM

Recall that in min-regularized-norm, the regularization on $\lambda$ can take various forms. In the following, we discuss an alternative regularization, i.e., L2-regularization $r(\boldsymbol{\lambda}, \boldsymbol{\lambda}_0) = \frac{1}{2}\|\boldsymbol{\lambda} - \boldsymbol{\lambda}_0\|_2^2$. In the discussion, we will show that similar to (Sener & Koltun, 2018), min-regularized-norm with L2-norm can be computed very efficiently via the Frank-Wolfe method.

Similar to the previous discussion on L1-regularization, we first consider the setting of $m = 2$. In this case, for any $\boldsymbol{\lambda} = (\gamma, 1 - \gamma), \boldsymbol{\lambda}_0 = (\gamma_0, 1 - \gamma_0) \in \mathcal{S}_2$, the L2-regularization $\frac{1}{2}\|\boldsymbol{\lambda} - \boldsymbol{\lambda}_0\|_2^2$ equals to $(\gamma - \gamma_0)^2$. Hence min-regularized-norm with L2-norm at round $t$ reduces to $\boldsymbol{\lambda}_t = (\gamma_t, 1 - \gamma_t)$ where

$$\gamma_t \in \arg\min_{0 \le \gamma \le 1} \|\gamma \boldsymbol{g}_1 + (1 - \gamma)\boldsymbol{g}_2\|_2^2 + \alpha(\gamma - \gamma_0)^2.$$

Since the above problem is in the quadratic form, it also has a closed-form solution. The proof is elementary and hence omitted.

**Proposition 4.** *Min-regularized-norm with L2-norm produces weights $\boldsymbol{\lambda}_t = (\gamma_t, 1 - \gamma_t)$ where*

$$\gamma_t = \max\{\min\{\frac{(\boldsymbol{g}_2 - \boldsymbol{g}_1)^\top \boldsymbol{g}_2 + \alpha\gamma_0}{\|\boldsymbol{g}_2 - \boldsymbol{g}_1\|_2^2 + \alpha}, 1\}, 0\}.$$

When $m > 2$, since $\boldsymbol{\lambda}_t$ is constrained to the probability simplex $\Delta_m$, similar to the case of L1-regularization, we can use a Frank-Wolfe method to efficiently calculate the composition weights, which is presented in Algorithm 3. Note that since the line search (step 2) has a closed-form solution, its calculation cost is not high, i.e., just the same as the calculation cost of the original min-norm solver (Sener & Koltun, 2018).

## F   MORE DETAILS OF THE THEORETICAL RESULTS

In this section, we first prove the remark below Theorem 1, i.e., with proper choices of $\eta_t$ and $\alpha_t$, DR-OMMD is guaranteed to have a sublinear regret bound in $O(\sqrt{T})$. Then we show the tightness of the above derived regret bound of DR-OMMD. Finally, we give a more detailed comparison with linearization from the theoretical aspect.

### F.1   MORE DETAILS OF THE REMARK BELOW THEOREM 1

Recall that in the remark below Theorem 1 in our main paper, we claim that with proper choice of $\eta_t$ and $\alpha_t$, DR-OMMD is guaranteed to attain a sublinear regret bound. We summarize this remark into the following corollary and provide a strict proof.

**Corollary 1.** *(i) (**Fixed learning rate**) When setting $\eta_t = \frac{\sqrt{2\gamma D}}{G\sqrt{T}}$ and $\alpha_t = \frac{4F}{\eta_t}$, for any $\boldsymbol{\lambda}_0 \in \mathcal{S}_m$, DR-OMMD achieves the following multi-objective regret*

$$R(T) \leq G\sqrt{2\gamma DT}.$$

*(ii) (**Diminishing learning rate**) When setting $\eta_t = \frac{\sqrt{2\gamma D}}{G\sqrt{t}}, \alpha_t = \frac{4F}{\eta_t}$, for any $\boldsymbol{\lambda}_0 \in \mathcal{S}_m$, DR-OMMD attains the following multi-objective regret*

$$R(T) \leq \frac{3}{2}G\sqrt{2\gamma DT}.$$

*Proof.* We start from the regret bound regarding $\boldsymbol{\lambda}_t$ in Theorem 1. When $\alpha_t = \frac{4F}{\eta_t}$, from the definition of min-regularized-norm, the composite weights $\lambda_t$ generated by DR-OMMD at each round satisfy

$$\boldsymbol{\lambda}_t \in \underset{\boldsymbol{\lambda} \in \mathcal{S}_m}{\arg\min} \|\nabla F_t(\boldsymbol{x}_t)\boldsymbol{\lambda}\|_2^2 + \frac{4F}{\eta_t}\|\boldsymbol{\lambda} - \boldsymbol{\lambda}_0\|_1.$$

Recall that $\boldsymbol{\lambda}_0 \in \mathcal{S}_m$. Hence, for any $t \in \{1, ..., T\}$, the last term of the regret bound in Theorem 1 can be bounded as

$$\|\nabla F_t(\boldsymbol{x}_t)\boldsymbol{\lambda}_t\|_2^2 + \frac{4F}{\eta_t}\|\boldsymbol{\lambda}_t - \boldsymbol{\lambda}_0\|_1 = \min_{\boldsymbol{\lambda} \in \mathcal{S}_m} \|\nabla F_t(\boldsymbol{x}_t)\boldsymbol{\lambda}\|_2^2 + \frac{4F}{\eta_t}\|\boldsymbol{\lambda} - \boldsymbol{\lambda}_0\|_1 \leq \|\nabla F_t(\boldsymbol{x}_t)\boldsymbol{\lambda}_0\|_2^2.$$

From Assumption 2, each gradient $\boldsymbol{g}_t^i$ is bounded as $\|\boldsymbol{g}_t^i\|_2 \leq G$, hence $\|\nabla F_t(\boldsymbol{x}_t)\boldsymbol{\lambda}_0\|_2 \leq \sum_{i=1}^m \|\lambda_0^i \boldsymbol{g}_t^i\|_2 = \sum_{i=1}^m \lambda_0^i \|\boldsymbol{g}_t^i\|_2 \leq G$. Therefore, when $\alpha_t = \frac{4F}{\eta_t}$, we have

$$\|\nabla F_t(\boldsymbol{x}_t)\boldsymbol{\lambda}_t\|_2^2 + \frac{4F}{\eta_t}\|\boldsymbol{\lambda}_t - \boldsymbol{\lambda}_0\|_1 \leq \|\nabla F_t(\boldsymbol{x}_t)\boldsymbol{\lambda}_0\|_2^2 \leq G^2.$$

Plugging it into the regret bound in Theorem 1, we have

$$R(T) \leq \frac{\gamma D}{\eta_T} + \frac{\eta_t}{2}G^2 T = G\sqrt{2\gamma DT},$$

which proves the bound with the fixed optimal learning rate.

Alternatively, set $\eta_t = \frac{\sqrt{2\gamma D}}{G\sqrt{t}}$ and utilize $\sum_{t=1}^T \frac{1}{\sqrt{t}} \leq 2\sqrt{T}$, we also have

$$R(T) \leq \frac{1}{2}G\sqrt{2\gamma DT}(1 + \sum_{t=1}^T \frac{1}{\sqrt{t}}) \leq \frac{3}{2}G\sqrt{2\gamma DT},$$

which proves the bound with the adaptive learning rate. ∎

## F.2 THE TIGHTNESS OF DR-OMMD'S BOUND

In this subsection, we show that the derived bound in Corollary 1 is tight w.r.t. $m$ regarding any gradient-based algorithm. Specifically, we follow the standard worst-case analysis of deriving lower bounds and construct a special case in which any gradient-based algorithm will incur a regret in the order of $\Omega(\sqrt{T})$.

Assume $f_t^1 = f_t^2 = \cdots = f_t^m$ at each round $t$. In this case, the instantaneous gradients of all the objectives are identical, i.e., $\boldsymbol{g}_t^i = \nabla f_t^i(\boldsymbol{x}_t) \equiv \nabla f_t^1(\boldsymbol{x}_t) = \boldsymbol{g}_t^1, \forall i \in \{1, ..., m\}$. For any gradient-based algorithm, since it can only utilize the gradient information of the objectives, it cannot distinguish the objective to which a certain gradient belongs. Alternatively speaking, in this case, any multiple gradient algorithm will treat all gradients in the same way and thus behave like a single-objective algorithm using the single gradient $\boldsymbol{g}_t^1$. Hence, in intuition, for any gradient-based algorithm, the worst-case bounds are at least independent of $m$. In particular, the worst-case bounds of gradient-based algorithms cannot decrease as $m$ increases; otherwise, the above case will be violated.

In the following, we provide a detailed proof of the tightness of the $O(\sqrt{T})$ bound. In the above case, since $f_t^1 = f_t^2 = \cdots = f_t^m$ for any $t$, the cumulative losses of all the objectives are also identical, i.e., $\sum_{t=1}^T f_t^1 = \sum_{t=1}^T f_t^2 = \cdots = \sum_{t=1}^T f_t^m$. Therefore, the Pareto set $\mathcal{X}^*$ of the cumulative vector loss $\sum_{t=1}^T F_t$ coincides with the optimal decision set of the cumulative loss $\sum_{t=1}^T f_t^1$ of the first objective, i.e., $\mathcal{X}^* = \operatorname{argmin}_{\boldsymbol{x} \in \mathcal{X}} \sum_{t=1}^T f_t^1(\boldsymbol{x})$. Recall our definition of the multi-objective regret. Since $\boldsymbol{\lambda}^\top F_t(\boldsymbol{x}) = f_t^1(\boldsymbol{x})$ for any $\boldsymbol{\lambda} \in \mathcal{S}_m$, we have

$$R(T) = \sup_{\boldsymbol{x}^* \in \mathcal{X}^*} \left( \sum_{t=1}^T f_t^1(\boldsymbol{x}_t) - \sum_{t=1}^T f_t^1(\boldsymbol{x}^*) \right) = \sum_{t=1}^T f_t^1(\boldsymbol{x}_t) - \min_{\boldsymbol{x}^* \in \mathcal{X}} \sum_{t=1}^T f_t^1(\boldsymbol{x}^*),$$

which exactly reduces to the single-objective regret $R_S(T)$ defined by the losses $\{f_t^1\}_{t=1}^T$ of the first objective. Hence we have $R(T) = R_S(T)$ in this case. Since the losses $\{f_t^1\}_{t=1}^T$ of the first objective can be chosen adversarially, we can follow Section 3.2 in (Hazan et al., 2016) to construct a certain sequence $\{f_t^1\}_{t=1}^T$ that admits a lower single-objective regret bound of $\Omega(\sqrt{T})$. Hence in this certain case, any multiple gradient algorithm will admit a multi-objective regret $R(T) = \Omega(\sqrt{T})$ w.r.t. $T$ and $m$, matching our derived regret bound for DR-OMMD in terms of both $T$ and $m$.

Some readers may suspect it unreasonable that in the multi-objective setting, the derived regret bounds do not increase as $m$ increases. Now we explicate the rationality of such independence in the following.

In fact, the independence of $m$ lies in the adoption of PSG in the formulation of the regret. Recall that, in the definition of PSG, "$\exists i \in \{1, \ldots, m\}$" means that it just needs to pick one coordinate $i$ to satisfy $f_t^i(\boldsymbol{x}_t) - \epsilon < f_t^i(\boldsymbol{x}'')$, which omits the dependency of $m$. We can see this point from another perspective. Recall that in the derivation of Proposition 1, we know that the regret $R(T)$ has an equivalent form, namely $\sup_{\boldsymbol{x}^* \in \mathcal{X}^*} \inf_{\boldsymbol{\lambda}^* \in \mathcal{S}_m} (\boldsymbol{\lambda}^*)^\top \sum_{t=1}^T (F_t(\boldsymbol{x}_t) - F_t(\boldsymbol{x}^*))$, or equivalently $\sup_{\boldsymbol{x}^* \in \mathcal{X}^*} \min_{i \in \{1, \ldots, m\}} \sum_{t=1}^T (f_t^i(\boldsymbol{x}_t) - f_t^i(\boldsymbol{x}^*))$. In particular, PSG takes a minimum operation over all objectives, and thus it does not necessarily increase as $m$ increases.

There is another intuitive way that can help understand the rationality of the independence of $m$. As is well recognized in existing research in multi-objective optimization (Emmerich & Deutz, 2018), the proportion of the Pareto optimal solutions (or more precisely, non-dominated solutions) in the decision domain tends to increase rapidly as the number of objectives increases. As a consequence, it might not be harder to reach the Pareto optimal set when $m$ turns larger, hence intuitively, the regret bound does not necessarily increase as $m$ increases.

## F.3 MORE DETAILS IN THE COMPARISON WITH LINEARIZATION

Recall that in the remark below Theorem 1, we show that our derived bound for DR-OMMD is smaller than that of linearization, and discuss the margin between the two regret bounds in the two-objective setting with linear losses. We now summarize the result in Theorem 2 in the following.

**Theorem 2.** *Consider a two-objective optimization setting with linear losses. Suppose the loss functions are $f_t^1(\boldsymbol{x}) = \boldsymbol{x}^\top \boldsymbol{g}_t^1$ and $f_t^2(\boldsymbol{x}) = \boldsymbol{x}^\top \boldsymbol{g}_t^2$ at each round $t$. For any $\boldsymbol{\lambda}_0 = (\gamma_0, 1 - \gamma_0) \in \mathcal{S}_m$,*

*let $\boldsymbol{\lambda}_t = (\gamma_t, 1 - \gamma_t)$ denote the composite weights produced by min-regularized-norm with L1-norm. When the regularization strength is set as $\alpha_t = 4F/\eta_t$, the margin between the regret bound of linearization with fixed weights $\boldsymbol{\lambda}_0$ and that of DR-OMMD with composite weights $\boldsymbol{\lambda}_t$ is at least*

$$M \geq \sum_{t=1}^{T} \frac{\eta_t}{2} (\gamma_t - \gamma_0)^2 \|\boldsymbol{g}_t^1 - \boldsymbol{g}_t^2\|_2^2.$$

Before proving the theorem, we remark that the two bounds are basically in the same order. Note that this theoretical result is also very commonly seen in the offline setting, where multiple gradient algorithms often have the same (convergence) rate as linearization (Yu et al., 2020; Liu et al., 2021). The benefit of multiple gradient algorithms is mainly due to the implementation of gradient composition. For example, the concept of common descent (Sener & Koltun, 2018; Yu et al., 2020) eliminates the gradient conflicting issue; the resulting algorithm achieves substantial performance improvements compared to linearization in their experiments. In this paper, we move one step forward and discuss the margin between DR-OMMD and linearization. We show that such a margin is due to the gradient difference $\boldsymbol{g}_t^1 - \boldsymbol{g}_t^2$ and the gap between the pre-defined weights $\boldsymbol{\lambda}_0$ and the adaptive weights $\boldsymbol{\lambda}_t$. This is the best we can do for now. The regret bound comparison for $m \geq 3$ is left for future research.

*Proof of Theorem 2.* We first write out the regret bounds of both methods. For DR-OMMD with $\alpha_t = 4F/\eta_t$, Theorem 1 provides the following regret bound

$$R_{\text{DR-OMMD}}(T) \leq \frac{\gamma D}{\eta_T} + \sum_{t=1}^{T} \frac{\eta_t}{2} \min_{\boldsymbol{\lambda} \in \mathcal{S}_m} \{\|\nabla F_t(\boldsymbol{x}_t)\boldsymbol{\lambda}\|_2^2 + \frac{4F}{\eta_t} \|\boldsymbol{\lambda} - \boldsymbol{\lambda}_0\|_1\}.$$

For linearization with fixed weights $\boldsymbol{\lambda}_0 \in \mathcal{S}_m$, it can be viewed as single-objective optimization with linearized loss $\boldsymbol{\lambda}_0^\top F_t$. Hence, we can directly borrow the tight bound of OMD (e.g., Theorem 6.8 in (Orabona, 2019)) and derive a bound

$$R_{\text{linear}}(T) \leq \frac{\gamma D}{\eta_T} + \sum_{t=1}^{T} \frac{\eta_t}{2} \|\nabla F_t(\boldsymbol{x}_t)\boldsymbol{\lambda}_0\|_2^2.$$

The margin between the above two bounds takes

$$M = \sum_{t=1}^{T} \frac{\eta_t}{2} (\|\nabla F_t(\boldsymbol{x}_t)\boldsymbol{\lambda}_0\|_2^2 - \min_{\boldsymbol{\lambda} \in \mathcal{S}_m} \{\|\nabla F_t(\boldsymbol{x}_t)\boldsymbol{\lambda}\|_2^2 + \frac{4F}{\eta_t} \|\boldsymbol{\lambda} - \boldsymbol{\lambda}_0\|_1\}),$$

which stems from different choices of composite weights. We investigate the margin at each round.

**Lemma 3.** *In a two-objective setting, suppose the gradients are $g_1$ and $g_2$ at some specific round $t$, and the corresponding gradient matrix $\mathbf{G} = [\boldsymbol{g}_1, \boldsymbol{g}_2]$. For any $\boldsymbol{\lambda}_0 = (\gamma_0, 1 - \gamma_0) \in \mathcal{S}_m$, let $\boldsymbol{\lambda}_t = (\gamma_t, 1 - \gamma_t)$ denote the composite weights produced by min-regularized-norm with L1-norm, then the following inequality holds, i.e.,*

$$\|\mathbf{G}\boldsymbol{\lambda}_0\|_2^2 - (\|\mathbf{G}\boldsymbol{\lambda}_t\|_2^2 + \alpha\|\boldsymbol{\lambda}_t - \boldsymbol{\lambda}_0\|_1) \geq (\gamma_0 - \gamma_t)^2 \|\boldsymbol{g}_2 - \boldsymbol{g}_1\|_2^2.$$

*Proof.* Denote the left side of the target inequality as $M(\boldsymbol{\lambda}_t, \boldsymbol{\lambda}_0)$, then it can be simplified as

$$M(\boldsymbol{\lambda}_t, \boldsymbol{\lambda}_0) = (\mathbf{G}\boldsymbol{\lambda}_0 - \mathbf{G}\boldsymbol{\lambda}_t)^\top (\mathbf{G}\boldsymbol{\lambda}_0 + \mathbf{G}\boldsymbol{\lambda}_t) - \alpha\|\boldsymbol{\lambda}_t - \boldsymbol{\lambda}_0\|_1$$
$$= (\boldsymbol{\lambda}_0 - \boldsymbol{\lambda}_t)^\top \mathbf{G}^\top \mathbf{G}(\boldsymbol{\lambda}_0 + \boldsymbol{\lambda}_t) - \alpha\|\boldsymbol{\lambda}_t - \boldsymbol{\lambda}_0\|_1$$

To leverage this term, we need to plug the derived composite weights $\boldsymbol{\lambda}_t$ into $M(\boldsymbol{\lambda}_t, \boldsymbol{\lambda}_0)$. Recall that in the two-objective setting, the weight $\gamma_t$ is given as

$$\gamma_t = \max\{\min\{\gamma_t'', 1\}, 0\}, \quad \text{where } \gamma_t'' = \max\{\min\{\gamma_0, \gamma_R\}, \gamma_L\},$$

where $\gamma_L = (\boldsymbol{g}_2^\top(\boldsymbol{g}_2 - \boldsymbol{g}_1) - \alpha)/\|\boldsymbol{g}_2 - \boldsymbol{g}_1\|_2^2$ and $\gamma_R = (\boldsymbol{g}_2^\top(\boldsymbol{g}_2 - \boldsymbol{g}_1) + \alpha)/\|\boldsymbol{g}_2 - \boldsymbol{g}_1\|_2^2$. Since the maximum and minimum operations will truncate the value of the produced weight, we now calculate $M(\boldsymbol{\lambda}_t, \boldsymbol{\lambda}_0)$ by case. Specifically, notice that $\gamma_L < \gamma_R$ and $0 \leq \gamma_0 \leq 1$, we consider the following cases.

**Case 1:** When $\gamma_R < 0$, we must have $\gamma_L < \gamma_R < 0 \leq \gamma_0$, which leads to $\gamma_t = 0$. In this case, we have $\boldsymbol{\lambda}_0 - \boldsymbol{\lambda}_t = (\gamma_0, -\gamma_0)$ and $\boldsymbol{\lambda}_0 + \boldsymbol{\lambda}_t = (\gamma_0, 2 - \gamma_0)$. Therefore, $M(\boldsymbol{\lambda}_t, \boldsymbol{\lambda}_0)$ can be computed as

$$M(\boldsymbol{\lambda}_t, \boldsymbol{\lambda}_0) = (\gamma_0 \boldsymbol{g}_1 - \gamma_0 \boldsymbol{g}_2)^\top (\gamma_0 \boldsymbol{g}_1 + (2 - \gamma_0)\boldsymbol{g}_2) - 2\alpha\boldsymbol{\lambda}_0.$$

Also, from the condition $\gamma_R < 0$, we have $\alpha < \boldsymbol{g}_2^\top (\boldsymbol{g}_1 - \boldsymbol{g}_2)$. Since $\boldsymbol{\lambda}_0 \geq 0$, plugging it into the above inequality, we have

$$M(\boldsymbol{\lambda}_t, \boldsymbol{\lambda}_0) \geq (\gamma_0 \boldsymbol{g}_1 - \gamma_0 \boldsymbol{g}_2)^\top (\gamma_0 \boldsymbol{g}_1 - \gamma_0 \boldsymbol{g}_2) = \gamma_0^2 \|\boldsymbol{g}_1 - \boldsymbol{g}_2\|_2^2.$$

In this case, since $\gamma_t = 0$, we derive the desired inequality.

**Case 2:** When $\gamma_L > 1$, we must have $\gamma_0 \leq 1 < \gamma_L < \gamma_R$, which results in $\gamma_t = 1$. In this case, we have $\boldsymbol{\lambda}_0 - \boldsymbol{\lambda}_t = (\gamma_0 - 1, 1 - \gamma_0)$ and $\boldsymbol{\lambda}_0 + \boldsymbol{\lambda}_t = (\gamma_0 + 1, 1 - \gamma_0)$. Now $M(\boldsymbol{\lambda}_t, \boldsymbol{\lambda}_0)$ can be calculated as

$$M(\boldsymbol{\lambda}_t, \boldsymbol{\lambda}_0) = ((\gamma_0 - 1)\boldsymbol{g}_1 + (1 - \gamma_0)\boldsymbol{g}_2)^\top ((\gamma_0 + 1)\boldsymbol{g}_1 + (1 - \gamma_0)\boldsymbol{g}_2) - 2\alpha(1 - \gamma_0).$$

Notice that the condition $\gamma_L > 1$ gives $\alpha < \boldsymbol{g}_1^\top (\boldsymbol{g}_2 - \boldsymbol{g}_1)$. Since $1 - \gamma_0 \geq 0$, plugging it into the above inequality, we have

$$M(\boldsymbol{\lambda}_t, \boldsymbol{\lambda}_0) \geq ((\gamma_0 - 1)\boldsymbol{g}_1 + (1 - \gamma_0)\boldsymbol{g}_2)^\top ((\gamma_0 - 1)\boldsymbol{g}_1 + (1 - \gamma_0)\boldsymbol{g}_2) = (1 - \gamma_0)^2 \|\boldsymbol{g}_1 - \boldsymbol{g}_2\|_2^2.$$

In this case, since $\gamma_t = 1$, we derive the desired inequality.

**Case 3:** When $0 \leq \gamma_L \leq \gamma_R \leq 1$, the margin is a bit more complex since the value of $\boldsymbol{\lambda}_t$ further depends on the relation between $\boldsymbol{\lambda}_0, \boldsymbol{\lambda}_L$, and $\boldsymbol{\lambda}_R$. Specifically, we consider the following cases.

(i) If $0 \leq \gamma_L \leq \gamma_0 \leq \gamma_R \leq 1$, then $\gamma_t = \gamma_0$. In this case, the inequality trivially holds.

(ii) If $0 \leq \gamma_0 \leq \gamma_L \leq 1$, then $\gamma_t = \gamma_L$. In this case, since $\|\boldsymbol{\lambda}_t - \boldsymbol{\lambda}_0\|_1 = 2(\gamma_L - \gamma_0)$, we can calculate $M(\boldsymbol{\lambda}_t, \boldsymbol{\lambda}_0)$ as

$$\begin{aligned} M(\boldsymbol{\lambda}_t, \boldsymbol{\lambda}_0) &= ((\gamma_0 - \gamma_L)\boldsymbol{g}_1 + (\gamma_L - \gamma_0)\boldsymbol{g}_2)^\top ((\gamma_0 + \gamma_L)\boldsymbol{g}_1 + (2 - \gamma_0 - \gamma_L)\boldsymbol{g}_2) - 2\alpha(\gamma_L - \gamma_0) \\ &= (\gamma_0 - \gamma_L)((\boldsymbol{g}_1 - \boldsymbol{g}_2)^\top ((\gamma_0 + \gamma_L)\boldsymbol{g}_1 + (2 - \gamma_0 - \gamma_L)\boldsymbol{g}_2) + 2\alpha). \end{aligned}$$

Since $\gamma_t = \gamma_L = (\boldsymbol{g}_2^\top (\boldsymbol{g}_2 - \boldsymbol{g}_1) - \alpha)/\|\boldsymbol{g}_2 - \boldsymbol{g}_1\|_2^2$, we have $\alpha = (\boldsymbol{g}_1 - \boldsymbol{g}_2)^\top (-\gamma_L \boldsymbol{g}_1 + (\gamma_L - 1)\boldsymbol{g}_2)$. Therefore, we further have

$$\begin{aligned} (\boldsymbol{g}_1 - \boldsymbol{g}_2)^\top ((\gamma_0 + \gamma_L)\boldsymbol{g}_1 + (2 - \gamma_0 - \gamma_L)\boldsymbol{g}_2) + 2\alpha &= (\boldsymbol{g}_1 - \boldsymbol{g}_2)^\top ((\gamma_0 - \gamma_L)\boldsymbol{g}_1 + (\gamma_L - \gamma_0)\boldsymbol{g}_2) \\ &= (\boldsymbol{g}_1 - \boldsymbol{g}_2)^\top (\gamma_0 - \gamma_L)(\boldsymbol{g}_1 - \boldsymbol{g}_2). \end{aligned}$$

Plugging it into the above equation on $M(\boldsymbol{\lambda}_t, \boldsymbol{\lambda}_0)$, we derive

$$M(\boldsymbol{\lambda}_t, \boldsymbol{\lambda}_0) = (\gamma_L - \gamma_0)^2 \|\boldsymbol{g}_1 - \boldsymbol{g}_2\|_2^2$$

(iii) If $0 \leq \gamma_R \leq \gamma_0 \leq 1$, then $\gamma_t = \gamma_R$. In this case, $\|\boldsymbol{\lambda}_t - \boldsymbol{\lambda}_0\|_1 = 2(\gamma_0 - \gamma_R)$, and $M(\boldsymbol{\lambda}_t, \boldsymbol{\lambda}_0)$ can be calculated as

$$\begin{aligned} M(\boldsymbol{\lambda}_t, \boldsymbol{\lambda}_0) &= ((\gamma_0 - \gamma_R)\boldsymbol{g}_1 + (\gamma_R - \gamma_0)\boldsymbol{g}_2)^\top ((\gamma_0 + \gamma_R)\boldsymbol{g}_1 + (2 - \gamma_0 - \gamma_R)\boldsymbol{g}_2) - 2\alpha(\gamma_0 - \gamma_R) \\ &= (\gamma_0 - \gamma_R)((\boldsymbol{g}_1 - \boldsymbol{g}_2)^\top ((\gamma_0 + \gamma_R)\boldsymbol{g}_1 + (2 - \gamma_0 - \gamma_R)\boldsymbol{g}_2) - 2\alpha). \end{aligned}$$

Since $\gamma_t = (\boldsymbol{g}_2^\top (\boldsymbol{g}_2 - \boldsymbol{g}_1) + \alpha)/\|\boldsymbol{g}_2 - \boldsymbol{g}_1\|_2^2$, we have $\alpha = (\boldsymbol{g}_1 - \boldsymbol{g}_2)^\top (\gamma_R \boldsymbol{g}_1 + (1 - \gamma_R)\boldsymbol{g}_2)$, then

$$(\boldsymbol{g}_1 - \boldsymbol{g}_2)^\top ((\gamma_0 + \gamma_R)\boldsymbol{g}_1 + (2 - \gamma_0 - \gamma_R)\boldsymbol{g}_2) - 2\alpha = (\boldsymbol{g}_1 - \boldsymbol{g}_2)^\top (\gamma_0 - \gamma_R)(\boldsymbol{g}_1 - \boldsymbol{g}_2).$$

Plugging it into the above equation on $M(\boldsymbol{\lambda}_t, \boldsymbol{\lambda}_0)$, we derive

$$M(\boldsymbol{\lambda}_t, \boldsymbol{\lambda}_0) = (\gamma_R - \gamma_0)^2 \|\boldsymbol{g}_1 - \boldsymbol{g}_2\|_2^2$$

Combining all of the above cases, we prove the lemma. ■

For any $t \in \{1, ..., T\}$, set $\boldsymbol{g}_1 = \boldsymbol{g}_t^1, \boldsymbol{g}_2 = \boldsymbol{g}_t^2$ (i.e., $\mathbf{G} = \mathbf{G}_t$), and $\alpha = \frac{4F}{\eta_t}$ in Lemma 3, we have

$$\|\mathbf{G}_t \boldsymbol{\lambda}_0\|_2^2 - (\|\mathbf{G}_t \boldsymbol{\lambda}_t\|_2^2 + \frac{4F}{\eta_t}\|\boldsymbol{\lambda}_t - \boldsymbol{\lambda}_0\|_1) \geq (\gamma_t - \gamma_0)^2 \|\boldsymbol{g}_t^2 - \boldsymbol{g}_t^1\|_2^2.$$

Since $\|\boldsymbol{\lambda}_t - \boldsymbol{\lambda}_0\|_2^2 = 2(\gamma_t - \gamma_0)^2$, summing the above inequality over $t \in \{1, ..., T\}$, we can directly calculate the margin as

$$M \geq \sum_{t=1}^T \frac{\eta_t}{4}\|\boldsymbol{\lambda}_t - \boldsymbol{\lambda}_0\|_2^2 \cdot \|\boldsymbol{g}_t^2 - \boldsymbol{g}_t^1\|_2^2,$$

which proves the theorem. ■

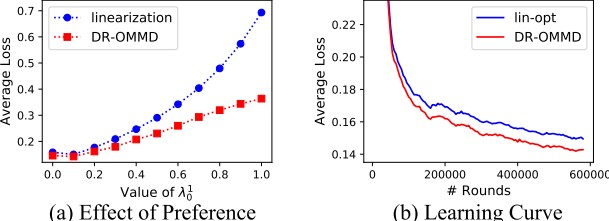

(a) Effect of Preference        (b) Learning Curve

Figure 3: Results to verify the effectiveness of adaptive regularization on *covtype*. (a) Performance of DR-OMMD and linearization under varying $\lambda_0 = (\lambda_0^1, 1 - \lambda_0^1)$. (b) Performance using the optimal weights $\lambda_0 = (0.1, 0.9)$.

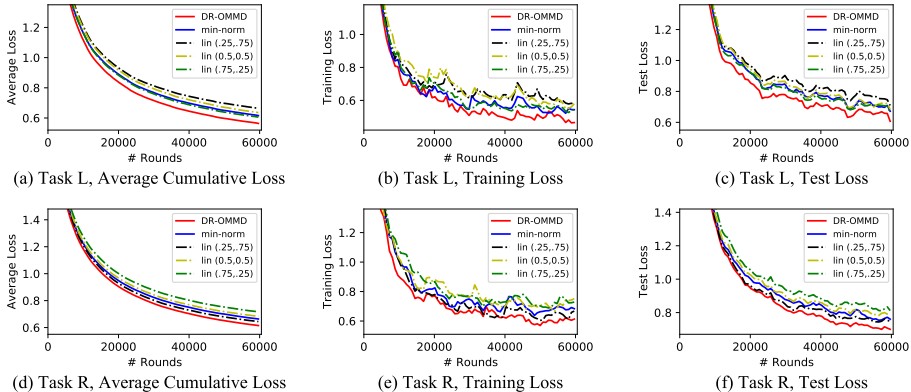

(a) Task L, Average Cumulative Loss    (b) Task L, Training Loss    (c) Task L, Test Loss

(d) Task R, Average Cumulative Loss    (e) Task R, Training Loss    (f) Task R, Test Loss

Figure 4: More empirical results of the multi-task deep online experiments on MultiMNIST. The plots show the average cumulative loss, training loss and test loss of DR-OMMD and various baselines on both tasks (*task L* and *task R*).

# G    MORE EXPERIMENTAL RESULTS

## G.1    MORE DETAILS OF THE EXPERIMENTAL SETUP

The *protein* and *covtype* datasets used in our experiments are publicly available in (Dua & Graff, 2017). The *MultiMNIST* dataset is acquired by the code provided by (Sener & Koltun, 2018).

All runs are deployed on Xeon(R) E5-2699 @ 2.2GHz.

## G.2    MORE RESULTS FOR ADAPTIVE REGULARIZATION

We supplement the empirical results on *covtype* in Figure 3, which have been omitted from our main paper due to the lack of space. These results are consistent with the results on *protein* as presented in our main paper.

## G.3    MORE RESULTS FOR ONLINE DEEP MULTI-TASK LEARNING

In our main paper, due to the page limit, Figure 2 only reports the average cumulative loss of DR-OMMD and various baselines on MultiMNIST. Here we supplement the results on the training loss and the test loss in Figure 4. The results are consistent with the average cumulative loss, showing superiority of DR-OMMD over linearization and MGDA in the non-convex setting.

# H    OMITTED PROOFS OF PROPOSITION 2 (MIN-NORM MAY INCUR LINEAR REGRETS)

*Proof.* As we have described in our main paper, we consider the following two-objective optimization problem. Decision domain is set as $\mathcal{X} = \{(u, v) \mid u + v \leq \frac{1}{2}, v - u \leq \frac{1}{2}, v \geq 0\}$. At each

round $t$, the loss function $F_t : \mathcal{X} \to \mathbb{R}^2$ takes

$$F_t(x) = \begin{cases} (\|\boldsymbol{x} - \boldsymbol{a}\|_2^2, \|\boldsymbol{x} - \boldsymbol{b}\|_2^2), & t = 2k - 1, k = 1, 2, ...; \\ (\|\boldsymbol{x} - \boldsymbol{b}\|_2^2, \|\boldsymbol{x} - \boldsymbol{c}\|_2^2), & t = 2k, k = 1, 2, ..., \end{cases}$$

where $\boldsymbol{a} = (-2, -1), \boldsymbol{b} = (0, 1), \boldsymbol{c} = (2, -1)$. For simplicity of analysis, we first consider the case when the total time horizon $T$ is an even number. Then it can be checked that the cumulative loss function takes

$$\sum_{t=1}^{T} F_t(\boldsymbol{x}) = \frac{T}{2} \cdot (\|\boldsymbol{x} - \boldsymbol{a}\|_2^2 + \|\boldsymbol{x} - \boldsymbol{b}\|_2^2, \|\boldsymbol{x} - \boldsymbol{b}\|_2^2 + \|\boldsymbol{x} - \boldsymbol{c}\|_2^2)$$

$$= T \cdot ((u+1)^2 + v^2 + 2, (u-1)^2 + v^2 + 2),$$

for any $\boldsymbol{x} = (u, v) \in \mathcal{X}$. Obviously the Pareto optimal set $\mathcal{X}^*$ of the cumulative loss coincides with the line segment between $(-1, 0)$ and $(1, 0)$, i.e., $\mathcal{X}^* = \{(u, v) \mid -\frac{1}{2} \le u \le \frac{1}{2}, v = 0\}$ (note that $\mathcal{X}^*$ is the intersection of the line segment and $\mathcal{X}$).

Now consider equipping OMD with vanilla min-norm, where the composite gradients are produced by the min-norm method. Suppose the learning process starts at any $\boldsymbol{x}_1 = (u_1, v_1) \in \mathcal{X}$ such that $v_1 > 0$. Note that this is true if and only if $\boldsymbol{x}_1 \notin \mathcal{X}^*$. Then for the iterate $\boldsymbol{x}_t = (u_t, v_t)$ at each round $t$, we can directly calculate the gradients as

$$\boldsymbol{g}_t^1 = \begin{cases} 2(\boldsymbol{x}_t - \boldsymbol{a}) = (2u_t + 4, & 2v_t + 2), & t = 2k - 1; \\ 2(\boldsymbol{x}_t - \boldsymbol{b}) = (2u_t, & 2v_t - 2), & t = 2k. \end{cases}$$

$$\boldsymbol{g}_t^2 = \begin{cases} 2(\boldsymbol{x}_t - \boldsymbol{b}) = (2u_t, & 2v_t - 2), & t = 2k - 1; \\ 2(\boldsymbol{x}_t - \boldsymbol{c}) = (2u_t - 4, & 2v_t + 2), & t = 2k. \end{cases}$$

The min-norm weights can be computed as $\boldsymbol{\lambda}_t = (\gamma_t, 1 - \gamma_t)$ where

$$\gamma_t = \begin{cases} \dfrac{(\boldsymbol{x}_t - \boldsymbol{b})^\top (\boldsymbol{a} - \boldsymbol{b})}{\|\boldsymbol{a} - \boldsymbol{b}\|_2^2} = \dfrac{1 - u_t - v_t}{4}, & t = 2k - 1; \\ \dfrac{(\boldsymbol{x}_t - \boldsymbol{c})^\top (\boldsymbol{b} - \boldsymbol{c})}{\|\boldsymbol{b} - \boldsymbol{c}\|_2^2} = \dfrac{3 - u_t + v_t}{4}, & t = 2k. \end{cases}$$

The composite gradient

$$\boldsymbol{g}_t^{comp} = \begin{cases} \gamma_t \cdot 2(\boldsymbol{x} - \boldsymbol{a}) + (1 - \gamma_t) \cdot 2(\boldsymbol{x} - \boldsymbol{b}) = (u_t - v_t + 1, & -u_t + v_t - 1), & t = 2k - 1; \\ \gamma_t \cdot 2(\boldsymbol{x} - \boldsymbol{b}) + (1 - \gamma_t) \cdot 2(\boldsymbol{x} - \boldsymbol{c}) = (-u_t - v_t - 1, & -u_t - v_t - 1), & t = 2k. \end{cases}$$

Recall that the update form of OMD takes

$$\boldsymbol{x}_{t+1} = \Pi_{\mathcal{X}}(\boldsymbol{x}_t - \eta_t \boldsymbol{g}_t^{comp}),$$

where $\eta_t > 0$ is the learning rate and $\Pi_{\mathcal{X}}$ is the projection operation onto $\mathcal{X}$. Denote the iterate $\boldsymbol{x}_t = (u_t, v_t)$ at each round. Now we can investigate the relation between $\boldsymbol{x}_t$ and $\boldsymbol{x}_{t+1}$ by considering the following two cases:

(i) If $\boldsymbol{x}_t - \eta_t \boldsymbol{g}_t^{comp} \in \mathcal{X}$, then we do not need projection, and directly have $\boldsymbol{x}_{t+1} = \boldsymbol{x}_t - \eta_t \boldsymbol{g}_t^{comp}$.

(ii) If $\boldsymbol{x}_t - \eta_t \boldsymbol{g}_t^{comp} \notin \mathcal{X}$, then we need to project $\boldsymbol{x}_t - \eta_t \boldsymbol{g}_t^{comp}$ back to $\mathcal{X}$. Denote $\boldsymbol{x}_{t+1}' = \boldsymbol{x}_t - \eta_t \boldsymbol{g}_t^{comp}$. For simplicity we consider the projection based on the Euclidean distance, namely $\Pi_{\mathcal{X}}(\boldsymbol{x}) = \arg\min_{\boldsymbol{x}' \in \mathcal{X}} \|\boldsymbol{x} - \boldsymbol{x}'\|_2^2$. Since the composite gradient is orthogonal to the boundary on which the iterate after projection $\boldsymbol{x}_{t+1} = \Pi_{\mathcal{X}}(\boldsymbol{x}_{t+1}')$ is located, it can be checked that $\boldsymbol{x}_{t+1}$ lies on the line segment linking $\boldsymbol{x}_t$ and $\boldsymbol{x}_{t+1}'$. Alternatively speaking, $\boldsymbol{x}_{t+1}$ can be expressed as $\boldsymbol{x}_t - \eta_t' \boldsymbol{g}_t^{comp}$ for some $0 \le \eta_t' < \eta_t$.

Combining the above two cases, we know that at each round $t$, there exists some $\eta_t' \in [0, \eta_t]$ such that $\boldsymbol{x}_{t+1} = \boldsymbol{x}_t - \eta_t' \boldsymbol{g}_t^{comp}$. Now we can analyze the relation between each entry of $\boldsymbol{x}_t$ and $\boldsymbol{x}_{t+1}$. Specifically, since the second entry of the composite gradient is always non-positive, namely $-u_t + v_t - 1 \le 0$ and $-u_t - v_t - 1 \le 0$, we have $v_{t+1} \ge v_t$ for any $t$. Moreover, since the first entry of $\boldsymbol{g}_t^{comp}$ is non-negative when $t = 2k - 1$, namely $u_{2k-1} - v_{2k-1} + 1 \ge 0$, we have $u_{2k} \le u_{2k-1}$ for

any $k$; since the first entry of $\boldsymbol{g}_t^{comp}$ is non-positive when $t = 2k$, namely $-u_{2k} - v_{2k} - 1 \le 0$, we have $u_{2k+1} \ge u_{2k}$ for any $k$.

Now we can go back to analyze the gap between the composite weights at any two consecutive rounds. It is easy to verify that $\gamma_{2k-1} < \gamma_{2k}$ and $\gamma_{2k} > \gamma_{2k+1}$, hence we have

$$\|\boldsymbol{\lambda}_{2k} - \boldsymbol{\lambda}_{2k-1}\|_1 = 2(\gamma_{2k} - \gamma_{2k-1}) = \frac{2 - (u_{2k} - u_{2k-1}) + (v_{2k} + v_{2k-1})}{2} \ge 1 + v_1,$$

$$\|\boldsymbol{\lambda}_{2k+1} - \boldsymbol{\lambda}_{2k}\|_1 = 2(\gamma_{2k} - \gamma_{2k+1}) = \frac{2 - (u_{2k} - u_{2k+1}) + (v_{2k} + v_{2k+1})}{2} \ge 1 + v_1.$$

Therefore, the composite weights $\boldsymbol{\lambda}_t$ indeed change radically at any two consecutive rounds.

The above analysis on $v_t$ also implies the failure of min-norm in this problem. Recall that any Pareto optimal solution $\boldsymbol{x}^* = (u^*, v^*) \in \mathcal{X}^*$ must satisfy $v^* = 0$. Suppose the initial iterate $\boldsymbol{x}_1 = (u_1, v_1)$ does not lie in $\mathcal{X}^*$, i.e., $v_1 > 0$, which is almost sure for random initialization $\boldsymbol{x}_1 \in \mathcal{X}$. Then we iteratively have $0 < v_1 \le v_2 \le ... \le v_T$, which means that $\boldsymbol{x}_t$ moves away from the Pareto set $\mathcal{X}^*$.

In the following, we strictly prove that min-norm indeed incurs a **linear** multi-objective regret. To calculate $R(T)$, we first investigate the quantity $R(\boldsymbol{x}^*, \boldsymbol{\lambda}) = \boldsymbol{\lambda}^\top \sum_{t=1}^T (F_t(\boldsymbol{x}_t) - F_t(\boldsymbol{x}^*))$ for any fixed weights $\boldsymbol{\lambda} = (\gamma, 1 - \gamma) \in \mathcal{S}_2$ and best fixed decision $\boldsymbol{x}^* = (u^*, 0) \in \mathcal{X}^*$. Specifically, recall the form of $\sum_{t=1}^T F_t$ derived above, then we have

$$\boldsymbol{\lambda}^\top \sum_{t=1}^T F_t(\boldsymbol{x}^*) = (\gamma(u^* + 1)^2 + (1 - \gamma)(u^* - 1)^2 + 2)T.$$

Denote the cumulative loss $\sum_{t=1}^T F_t(\boldsymbol{x}_t) = (L_1, L_2)$, we now consider the loss of each objective $L_1$ and $L_2$ separately. Specifically, for the first objective, we have

$$L_1 = \sum_{k=1}^{T/2} ((u_{2k-1} + 2)^2 + u_{2k}^2 + (v_{2k-1} + 1)^2 + (v_{2k} - 1)^2).$$

Since $0 < v_1 \le v_2 \le ... \le v_T \le 1$, for the term regarding $v_t$ we have

$$\sum_{k=1}^{T/2} ((v_{2k-1} + 1)^2 + (v_{2k} - 1)^2) = (v_1 + 1)^2 + (v_T - 1)^2 + \sum_{k=1}^{T/2-1} ((v_{2k} - 1)^2 + (v_{2k+1} + 1)^2)$$

$$\ge \sum_{k=1}^{T/2-1} ((v_{2k} - 1)^2 + (v_{2k} + 1)^2) = \sum_{k=1}^{T/2-1} (2v_{2k}^2 + 2)$$

$$\ge \sum_{k=1}^{T/2-1} (2v_1^2 + 2) = (2v_1^2 + 2)(\frac{T}{2} - 1) \ge v_1^2 T + T - 2.$$

For the $k$-th term regarding $u_t$, we have

$$(u_{2k-1} + 2)^2 + u_{2k}^2 = (u_{2k-1} + 1)^2 + (u_{2k} + 1)^2 + 2(u_{2k-1} - u_{2k}) + 2.$$

Recall that we have derived $u_{2k} \le u_{2k-1}$, thus we have

$$\sum_{k=1}^{T/2} (u_{2k-1} + 2)^2 + u_{2k}^2 \ge \sum_{k=1}^{T/2} ((u_{2k-1} + 1)^2 + (u_{2k} + 1)^2 + 2) \ge \sum_{t=1}^T (u_t + 1)^2 + T \ge (\bar{u} + 1)^2 T + T,$$

where $\bar{u} = \frac{1}{T} \sum_{t=1}^T u_t$ and the last inequality is derived from Jensen's inequality. In summary, for the cumulative loss $L_1$ of the first objective, we have

$$L_1 \ge (\bar{u} + 1)^2 T + v_1^2 T + 2T - 2.$$

Similarly, we can analyze the cumulative loss $L_2$ of the second objective

$$L_2 = \sum_{k=1}^{T/2} (u_{2k-1}^2 + (u_{2k} - 2)^2 + (v_{2k-1} - 1)^2 + (v_{2k} + 1)^2).$$

Since $0 < v_1 \leq v_2 \leq ... \leq v_T \leq 1$, for the term regarding $v_t$ we have

$$\sum_{k=1}^{T/2}((v_{2k-1}-1)^2 + (v_{2k}+1)^2) \geq \sum_{k=1}^{T/2}((v_{2k-1}-1)^2 + (v_{2k-1}+1)^2) \geq v_1^2 T + T.$$

For the term regarding $u_t$, we also have

$$\sum_{k=1}^{T/2}(u_{2k-1}^2 + (u_{2k}-2)^2) = \sum_{k=1}^{T/2}((u_{2k-1}-1)^2 + (u_{2k}-1)^2 + 2(u_{2k-1}-u_{2k}) + 2)$$

$$\geq \sum_{t=1}^{T}(u_t-1)^2 + T \geq (\bar{u}-1)^2 T + T,$$

where the last inequality is derived from Jensen's inequality. Therefore, we have

$$L_2 \geq (\bar{u}-1)^2 T + v_1^2 T + 2T.$$

Combining the above inequalities, we have

$$R(\boldsymbol{x}^*, \boldsymbol{\lambda}) = \gamma L_1 + (1-\gamma)L_2 - \boldsymbol{\lambda}^\top \sum_{t=1}^{T} F_t(\boldsymbol{x}^*)$$

$$\geq \gamma((\bar{u}+1)^2 - (u^*+1)^2) + (1-\gamma)((\bar{u}-1)^2 - (u^*-1)^2) + v_1^2 T - 2\gamma.$$

For any $\boldsymbol{\lambda} \in \mathcal{S}_2$ (i.e., $\gamma \in [0,1]$), set $\boldsymbol{x}' = (\bar{u}, 0) \in \mathcal{X}^*$, then it holds that

$$R(\boldsymbol{x}', \boldsymbol{\lambda}) \geq v_1^2 T - 2.$$

Equivalently, the multi-objective regret satisfies

$$R(T) = \sup_{\boldsymbol{x}^* \in \mathcal{X}^*} \inf_{\boldsymbol{\lambda} \in \mathcal{S}_2} R(\boldsymbol{x}^*, \boldsymbol{\lambda}) \geq \inf_{\boldsymbol{\lambda} \in \mathcal{S}_2} R(\boldsymbol{x}', \boldsymbol{\lambda}) \geq v_1^2 T - 2,$$

which is linear w.r.t. $T$ for any $\boldsymbol{x}_1 = (u_1, v_1) \in \mathcal{X}$ such that $v_1 > 0$.

We now investigate the case when $T$ is an odd number. Since the calculation of the composite weights $\boldsymbol{\lambda}_t$ and the composite gradient $\boldsymbol{g}_t^{comp}$ at each round is independent of the total time horizon $T$, we still have $\|\boldsymbol{\lambda}_{t+1} - \boldsymbol{\lambda}_t\|_1 \geq v_1 + 1$ for any $t$. Hence the first desired property also holds for any odd $T$.

It remains to prove that OMD with min-norm still incurs a linear regret when $T$ is odd. In this case, the Pareto optimal set $\mathcal{X}^*$ does not lie in the $x$-axis anymore, hence it is difficult to directly compute $R(T)$. However, we can still use our derived $R(\boldsymbol{x}^*, \boldsymbol{\lambda})$ for any even $T$ to estimate the regret. Specifically, set $\boldsymbol{x}' = (\frac{1}{T-1}\sum_{t=1}^{T-1} u_t, 0)$; from the above derivation with even $T$, for any $\boldsymbol{\lambda} \in \mathcal{S}_2$, we still have (note that now $T-1$ is an even number)

$$\boldsymbol{\lambda}^\top \sum_{t=1}^{T-1} F_t(\boldsymbol{x}_t) - \boldsymbol{\lambda}^\top \sum_{t=1}^{T-1} F_t(\boldsymbol{x}') \geq v_1^2 T - 2.$$

Since for any $\boldsymbol{x} \in \mathcal{X}$, we have $0 \leq \|\boldsymbol{x}-\boldsymbol{a}\|_2^2, \|\boldsymbol{x}-\boldsymbol{b}\|_2^2, \|\boldsymbol{x}-\boldsymbol{c}\|_2^2 \leq 10$, we have

$$R(\boldsymbol{x}', \boldsymbol{\lambda}) = \boldsymbol{\lambda}^\top \sum_{t=1}^{T} F_t(\boldsymbol{x}_t) - \boldsymbol{\lambda}^\top \sum_{t=1}^{T} F_t(\boldsymbol{x}') \geq v_1^2 T - 12.$$

Furthermore, from the definition of Pareto optimality, there exists some $\boldsymbol{x}'' \in \mathcal{X}^*$ that Pareto dominates $\boldsymbol{x}'$ regarding the cumulative loss $\sum_{t=1}^{T} F_t$, namely $\sum_{t=1}^{T} F_t(\boldsymbol{x}'') \preceq \sum_{t=1}^{T} F_t(\boldsymbol{x}')$. Hence

$$R(\boldsymbol{x}'', \boldsymbol{\lambda}) = \boldsymbol{\lambda}^\top \sum_{t=1}^{T} F_t(\boldsymbol{x}_t) - \boldsymbol{\lambda}^\top \sum_{t=1}^{T} F_t(\boldsymbol{x}'') \geq R(\boldsymbol{x}', \boldsymbol{\lambda}),$$

for any $\boldsymbol{\lambda} \in \mathcal{S}_2$. Therefore, the multi-objective regret

$$R(T) = \sup_{\boldsymbol{x}^* \in \mathcal{X}^*} \inf_{\boldsymbol{\lambda} \in \mathcal{S}_2} R(\boldsymbol{x}^*, \boldsymbol{\lambda}) \geq \inf_{\boldsymbol{\lambda} \in \mathcal{S}_2} R(\boldsymbol{x}'', \boldsymbol{\lambda}) \geq \inf_{\boldsymbol{\lambda} \in \mathcal{S}_2} R(\boldsymbol{x}', \boldsymbol{\lambda}) \geq v_1^2 T - 12,$$

which is also linear w.r.t. $T$ for any $\boldsymbol{x}_1 = (u_1, v_1) \in \mathcal{X}$ such that $v_1 > 0$. ∎

## I   OMITTED PROOFS OF THEOREM 1

*Proof.* We start from the definition of the multi-objective regret $R_{\mathrm{II}}(T)$ (which is abbreviated as $R(T)$). Specifically, for any $\boldsymbol{\lambda} \in \mathcal{S}_m$ and $\boldsymbol{\lambda}_1 \ldots, \boldsymbol{\lambda}_T \in \mathcal{S}_m$, it holds that

$$R(T) = \sup_{\boldsymbol{x}^* \in \mathcal{X}^*} \inf_{\boldsymbol{\lambda}^* \in \mathcal{S}_m} \sum_{t=1}^{T} \boldsymbol{\lambda}^{*\top}(F_t(\boldsymbol{x}_t) - F_t(\boldsymbol{x}^*)) \leq \sup_{\boldsymbol{x}^* \in \mathcal{X}^*} \sum_{t=1}^{T} \boldsymbol{\lambda}^{\top}(F_t(\boldsymbol{x}_t) - F_t(\boldsymbol{x}^*))$$

$$= \sup_{\boldsymbol{x}^* \in \mathcal{X}^*} \sum_{t=1}^{T} \left( (\boldsymbol{\lambda} - \boldsymbol{\lambda}_t)^{\top} F_t(\boldsymbol{x}_t) + \boldsymbol{\lambda}_t^{\top}(F_t(\boldsymbol{x}_t) - F_t(\boldsymbol{x}^*)) + (\boldsymbol{\lambda}_t - \boldsymbol{\lambda})^{\top} F_t(\boldsymbol{x}^*) \right)$$

$$\leq \sum_{t=1}^{T} F\|\boldsymbol{\lambda} - \boldsymbol{\lambda}_t\|_1 + \sup_{\boldsymbol{x}^* \in \mathcal{X}^*} \sum_{t=1}^{T} \boldsymbol{\lambda}_t^{\top}(F_t(\boldsymbol{x}_t) - F_t(\boldsymbol{x}^*)) + \sum_{t=1}^{T} F\|\boldsymbol{\lambda} - \boldsymbol{\lambda}_t\|_1$$

$$= 2F \sum_{t=1}^{T} \|\boldsymbol{\lambda} - \boldsymbol{\lambda}_t\|_1 + \sup_{\boldsymbol{x}^* \in \mathcal{X}^*} \sum_{t=1}^{T} \boldsymbol{\lambda}_t^{\top}(F_t(\boldsymbol{x}_t) - F_t(\boldsymbol{x}^*)).$$

To proceed, notice that if the composite weights $\boldsymbol{\lambda}_t$ are given beforehand instead of being calculated via min-regularized-norm, DR-OMMD acts just like standard OMD using linearized loss $\boldsymbol{\lambda}_t F_t$. Hence, the second term in the above regret bound can be further analyzed in a similar way as single-objective OMD (Srebro et al., 2011; Cesa-bianchi et al., 2012). Specifically, at each round $t$, since $F_t$ is coordinate-wise convex, the linearized loss $\boldsymbol{\lambda}_t^{\top} F_t$ is also convex. Also notice that the composite gradient $\boldsymbol{g}_t = \nabla F_t(\boldsymbol{x}_t)\boldsymbol{\lambda}_t$ is exactly the gradient of $\boldsymbol{\lambda}_t^{\top} F_t$ at $\boldsymbol{x}_t$. Hence for any $\boldsymbol{x}^* \in \mathcal{X}^*$, we have

$$\boldsymbol{\lambda}_t^{\top} F_t(\boldsymbol{x}_t) - \boldsymbol{\lambda}_t^{\top} F_t(\boldsymbol{x}^*) \leq \boldsymbol{g}_t^{\top}(\boldsymbol{x}_t - \boldsymbol{x}^*) = \boldsymbol{g}_t^{\top}(\boldsymbol{x}_{t+1} - \boldsymbol{x}^*) + \boldsymbol{g}_t^{\top}(\boldsymbol{x}_t - \boldsymbol{x}_{t+1}).$$

From the first-order optimal condition of $\boldsymbol{x}_{t+1}$, for any $\boldsymbol{x}' \in \mathcal{X}$, we have

$$(\eta_t \nabla F_t(\boldsymbol{x}_t)\boldsymbol{\lambda}_t + \nabla R(\boldsymbol{x}_{t+1}) - \nabla R(\boldsymbol{x}_t))^{\top}(\boldsymbol{x}' - \boldsymbol{x}_{t+1}) \geq 0.$$

Recall that $\boldsymbol{g}_t = \nabla F_t(\boldsymbol{x}_t)\boldsymbol{\lambda}_t$. We set $\boldsymbol{x}' = \boldsymbol{x}^*$ and combine the above two inequalities, which derives

$$\boldsymbol{\lambda}_t^{\top} F_t(\boldsymbol{x}_t) - \boldsymbol{\lambda}_t^{\top} F_t(\boldsymbol{x}^*) \leq \frac{1}{\eta_t}(\nabla R(\boldsymbol{x}_{t+1}) - \nabla R(\boldsymbol{x}_t))^{\top}(\boldsymbol{x}^* - \boldsymbol{x}_{t+1}) + \boldsymbol{g}_t^{\top}(\boldsymbol{x}_t - \boldsymbol{x}_{t+1}).$$

Recall the definition of Bregman divergence $B_R$. We can check that (also see (Beck & Teboulle, 2003))

$$B_R(\boldsymbol{x}^*, \boldsymbol{x}_t) - B_R(\boldsymbol{x}^*, \boldsymbol{x}_{t+1}) - B_R(\boldsymbol{x}_{t+1}, \boldsymbol{x}_t) = (\nabla R(\boldsymbol{x}_{t+1}) - \nabla R(\boldsymbol{x}_t))^{\top}(\boldsymbol{x}^* - \boldsymbol{x}_{t+1}).$$

Since $R$ is 1-strongly convex, we have $B_R(\boldsymbol{x}_{t+1}, \boldsymbol{x}_t) \geq \|\boldsymbol{x}_{t+1} - \boldsymbol{x}_t\|_2^2/2$. Hence

$$\boldsymbol{\lambda}_t^{\top} F_t(\boldsymbol{x}_t) - \boldsymbol{\lambda}_t^{\top} F_t(\boldsymbol{x}^*) \leq \frac{1}{\eta_t}(B_R(\boldsymbol{x}^*, \boldsymbol{x}_t) - B_R(\boldsymbol{x}^*, \boldsymbol{x}_{t+1}) - \frac{1}{2}\|\boldsymbol{x}_{t+1} - \boldsymbol{x}_t\|_2^2) + \boldsymbol{g}_t^{\top}(\boldsymbol{x}_t - \boldsymbol{x}_{t+1}).$$

Moreover, from the Cauchy-Schwartz inequality we have

$$\boldsymbol{g}_t^{\top}(\boldsymbol{x}_t - \boldsymbol{x}_{t+1}) \leq \frac{\eta_t}{2}\|\boldsymbol{g}_t\|_2^2 + \frac{1}{2\eta_t}\|\boldsymbol{x}_t - \boldsymbol{x}_{t+1}\|_2^2.$$

Combining the above two inequalities, we derive

$$\boldsymbol{\lambda}_t^{\top} F_t(\boldsymbol{x}_t) - \boldsymbol{\lambda}_t^{\top} F_t(\boldsymbol{x}^*) \leq \frac{1}{\eta_t}(B_R(\boldsymbol{x}^*; \boldsymbol{x}_t) - B_R(\boldsymbol{x}^*; \boldsymbol{x}_{t+1})) + \frac{\eta_t}{2}\|\nabla F_t(\boldsymbol{x}_t)\boldsymbol{\lambda}_t\|_2^2,$$

for any $\boldsymbol{x}^* \in \mathcal{X}^*$. Summing it over $t \in \{1, ..., T\}$ and utilizing $B_R(\boldsymbol{x}^*; \boldsymbol{x}_{T+1}) \geq 0$, we have

$$\sum_{t=1}^{T} \boldsymbol{\lambda}_t^{\top}(F_t(\boldsymbol{x}_t) - F_t(\boldsymbol{x}^*)) \leq \frac{1}{\eta_1} B_R(\boldsymbol{x}^*; \boldsymbol{x}_1) + \sum_{t=2}^{T}(\frac{1}{\eta_t} - \frac{1}{\eta_{t-1}}) B_R(\boldsymbol{x}^*; \boldsymbol{x}_T) + \sum_{t=1}^{T} \frac{\eta_t}{2}\|\nabla F_t(\boldsymbol{x}_t)\boldsymbol{\lambda}_t\|_2^2.$$

Since $B_R(\boldsymbol{x}^*; \boldsymbol{x}_t) \leq \gamma D$ and $\eta_t \leq \eta_{t-1}$ for any $t$, we have $(\frac{1}{\eta_t} - \frac{1}{\eta_{t-1}}) B_R(\boldsymbol{x}^*; \boldsymbol{x}_T) \leq (\frac{1}{\eta_t} - \frac{1}{\eta_{t-1}})\gamma D$. Hence we further have

$$\sum_{t=1}^{T} \boldsymbol{\lambda}_t^{\top}(F_t(\boldsymbol{x}_t) - F_t(\boldsymbol{x}^*)) \leq \frac{\gamma D}{\eta_T} + \sum_{t=1}^{T} \frac{\eta_t}{2}\|\nabla F_t(\boldsymbol{x}_t)\boldsymbol{\lambda}_t\|_2^2.$$

Taking the supremum over $\boldsymbol{x}^* \in \mathcal{X}^*$ and plugging it back to the above regret bound, we prove the theorem. ∎

## J    REGRET ANALYSIS IN THE STRONGLY CONVEX SETTING

In this section, we discuss the regret bound of DR-OMMD in the strongly convex setting, where each loss function $f_t^i$ is $H$-strongly convex. Recall that most literature in this setting only considers OGD (Zhao & Zhang, 2021; Wan et al., 2022), which is a special case of OMD that instantiates the regularization function on the iterate $\boldsymbol{x}$ as L2-regularizer, i.e., $R(\boldsymbol{x}) = \frac{1}{2}\|\boldsymbol{x}\|_2^2$. Hence, in the following, we mainly analyze the bound of the OGD-type variant in the strongly convex setting.

**Theorem 3.** *Assume that for any $t \in \{1, ..., T\}, i \in \{1, ..., m\}$, the loss function $f_t^i$ is $H$-strongly convex. Set $\eta_t = \frac{1}{Ht}$ and $R(\boldsymbol{x}) = \frac{1}{2}\|\boldsymbol{x}\|_2$ in DR-OMMD, then it attains the following regret*

$$R(T) \le \frac{H}{2}\|\boldsymbol{x}_1 - \boldsymbol{x}^*\|_2^2 + \sum_{t=1}^{T}\frac{1}{2Ht}(\|\nabla F_t(\boldsymbol{x}_t)\boldsymbol{\lambda}_t\|_2^2 + 4FHt\|\boldsymbol{\lambda}_t - \boldsymbol{\lambda}_0\|_1).$$

*Remark.* By setting $\alpha_t = 4FHt$, the above bound reduces to

$$R(T) \le \frac{H}{2}\|\boldsymbol{x}_1 - \boldsymbol{x}^*\|_2^2 + \sum_{t=1}^{T}\frac{1}{2Ht}\min_{\boldsymbol{\lambda}\in\mathcal{S}_m}\{\|\nabla F_t(\boldsymbol{x}_t)\boldsymbol{\lambda}\|_2^2 + \alpha_t\|\boldsymbol{\lambda} - \boldsymbol{\lambda}_0\|_1\}$$

$$\le \frac{H}{2}\|\boldsymbol{x}_1 - \boldsymbol{x}^*\|_2^2 + \sum_{t=1}^{T}\frac{\|\nabla F_t(\boldsymbol{x}_t)\boldsymbol{\lambda}_0\|_2^2}{2Ht} \le \frac{H}{2}\|\boldsymbol{x}_1 - \boldsymbol{x}^*\|_2^2 + \frac{G^2}{2H}\sum_{t=1}^{T}\frac{1}{t} = O(\log T),$$

which aligns with the optimal regret bound $O(\log T)$ in the single-objective strongly convex setting (Hazan et al., 2016).

*Proof.*  From the derivation of Theorem 1, we have

$$R(T) \le 2F\sum_{t=1}^{T}\|\boldsymbol{\lambda} - \boldsymbol{\lambda}_t\|_1 + \sup_{\boldsymbol{x}^*\in\mathcal{X}^*}\sum_{t=1}^{T}\boldsymbol{\lambda}_t^\top(F_t(\boldsymbol{x}_t) - F_t(\boldsymbol{x}^*)).$$

Denote the composite gradient as $\boldsymbol{g}_t = \nabla F_t(\boldsymbol{x}_t)\boldsymbol{\lambda}_t$, which equals to the gradient of $\boldsymbol{\lambda}_t F_t$ at $\boldsymbol{x}_t$. Since $\boldsymbol{\lambda}_t \in \mathcal{S}_m$, $\boldsymbol{\lambda}_t F_t$ is a convex combination of $f_t^1, ..., f_t^m$, hence $\boldsymbol{\lambda}_t F_t$ is also $H$-strongly convex. For any $\boldsymbol{x}^* \in \mathcal{X}^*$, we now have

$$\boldsymbol{\lambda}_t^\top F_t(\boldsymbol{x}_t) - \boldsymbol{\lambda}_t^\top F_t(\boldsymbol{x}^*) \le \boldsymbol{g}_t^\top(\boldsymbol{x}_t - \boldsymbol{x}^*) - \frac{H}{2}\|\boldsymbol{x}_t - \boldsymbol{x}^*\|_2^2.$$

We now bound the term $\boldsymbol{g}_t^\top(\boldsymbol{x}_t - \boldsymbol{x}^\star)$. When $R(\boldsymbol{x}) = \frac{1}{2}\|\boldsymbol{x}\|_2^2$, the Bregman divergence $B_R(\boldsymbol{x}, \boldsymbol{z}) = \frac{1}{2}\|\boldsymbol{x} - \boldsymbol{z}\|_2^2$ and the OMD update reduces to OGD, i.e., $\boldsymbol{x}_{t+1} = \Pi_\mathcal{X}(\boldsymbol{x}_t - \eta_t\boldsymbol{g}_t)$ ($\Pi_\mathcal{X}$ denotes the standard projection onto $\mathcal{X}$). Plugging the above update rule into $\|\boldsymbol{x}_{t+1} - \boldsymbol{x}^*\|_2^2$, we derive

$$\|\boldsymbol{x}_{t+1} - \boldsymbol{x}^\star\|_2^2 = \|\Pi_\mathcal{X}(\boldsymbol{x}_t - \eta_t\boldsymbol{g}_t) - \boldsymbol{x}^\star\|_2^2 \le \|\boldsymbol{x}_t - \eta_t\boldsymbol{g}_t - \boldsymbol{x}^\star\|_2^2,$$

where the last inequality is derived from the Pythagorean theorem. Hence we have

$$\|\boldsymbol{x}_{t+1} - \boldsymbol{x}^\star\|_2^2 \le \|\boldsymbol{x}_t - \boldsymbol{x}^\star\|_2^2 + \eta_t^2\|\boldsymbol{g}_t\|_2^2 - 2\eta_t\boldsymbol{g}_t^\top(\boldsymbol{x}_t - \boldsymbol{x}^\star),$$

or equivalently

$$\boldsymbol{g}_t^\top(\boldsymbol{x}_t - \boldsymbol{x}^\star) \le \frac{\|\boldsymbol{x}_t - \boldsymbol{x}^\star\|_2^2 - \|\boldsymbol{x}_{t+1} - \boldsymbol{x}^\star\|_2^2}{2\eta_t} + \frac{\eta_t}{2}\|\boldsymbol{g}_t\|_2^2.$$

Plugging it into the inequality of $\boldsymbol{\lambda}_t^\top F_t(\boldsymbol{x}_t) - \boldsymbol{\lambda}_t^\top F_t(\boldsymbol{x}^*)$, we derive

$$\boldsymbol{\lambda}_t^\top F_t(\boldsymbol{x}_t) - \boldsymbol{\lambda}_t^\top F_t(\boldsymbol{x}^*) \le \frac{\|\boldsymbol{x}_t - \boldsymbol{x}^\star\|_2^2 - \|\boldsymbol{x}_{t+1} - \boldsymbol{x}^\star\|_2^2}{2\eta_t} + \frac{\eta_t}{2}\|\boldsymbol{g}_t\|_2^2 - \frac{H}{2}\|\boldsymbol{x}_t - \boldsymbol{x}^*\|_2^2,$$

for any $\boldsymbol{x}^* \in \mathcal{X}^*$. Summing it over $t \in \{1, ..., T\}$, we have

$$\sum_{t=1}^{T}\boldsymbol{\lambda}_t^\top(F_t(\boldsymbol{x}_t) - F_t(\boldsymbol{x}^*))$$

$$\le \frac{1}{2\eta_1}\|\boldsymbol{x}_1 - \boldsymbol{x}^*\|_2^2 + \sum_{t=2}^{T}(\frac{1}{2\eta_t} - \frac{1}{2\eta_{t-1}} - \frac{H}{2})\|\boldsymbol{x}_t - \boldsymbol{x}^*\|_2^2 + \sum_{t=1}^{T}\frac{\eta_t}{2}\|\nabla F_t(\boldsymbol{x}_t)\boldsymbol{\lambda}_t\|_2^2.$$

When setting $\eta_t = 1/Ht$, we have

$$\frac{1}{2\eta_t} - \frac{1}{2\eta_{t-1}} - \frac{H}{2} = \frac{Ht}{2} - \frac{H(t-1)}{2} - \frac{H}{2} = 0.$$

Consequently, we have

$$\sum_{t=1}^{T} \boldsymbol{\lambda}_t^{\top} (F_t(\boldsymbol{x}_t) - F_t(\boldsymbol{x}^*)) \leq \frac{H}{2} \|\boldsymbol{x}_1 - \boldsymbol{x}^*\|_2^2 + \sum_{t=1}^{T} \frac{1}{2Ht} \|\nabla F_t(\boldsymbol{x}_t)\boldsymbol{\lambda}_t\|_2^2.$$

Plugging it into the above regret form, we derive the desired regret bound. ∎

