# OpenReview forum: "Multi-Objective Online Learning"
_ICLR.cc/2023/Conference — ICLR 2023 notable top 25%_

### Official Review · Reviewer_rnNt · 2022-10-25

**Confidence:** 3
**Correctness:** 3
**Technical Novelty And Significance:** 3
**Empirical Novelty And Significance:** 3
**Recommendation:** 5

**Clarity, Quality, Novelty And Reproducibility:**

The paper is written clearly in the main part, but the proof of the main result seems to be ambiguous to follow. With counterexamples, it provides the necessity to involve the sequence-wise PSG and min-regularized-norm. Under these two concepts, the paper gives a learnable algorithm for multi-objective online learning. The result now for m=2 is clear, but for the further result, it might not be convincing. The idea of this paper is acceptable and enlightening if the proof is correct.

**Strength And Weaknesses:**

This paper provides a new and efficient measurement of Multi-objective online learning. This measurement sequence-wise PSG is an extension of the original regret for the single loss function. Then, by proposing the algorithm, this paper involves the min-regularized norm to overcome the potential instability caused by the gradients for loss functions shown in a counterexample. This is a contribution.
However, there are some concerns about this paper.
1.	In the definition of sequence-wise PSG, the domain of the comparator of the algorithm is constrained to the Pareto optimality, not the total domain of the algorithm. Although the paper states the reason, it seems to be a very strong assumption of the problem. In one of the related works like “Online Minimax Multiobjective Optimization: Multicalibeating and Other Applications ” (D. Lee et al.), the comparator term is also constrained but the domain to the comparator term remains. This paper might be supposed to emphasize more in detail to support the advantages of this measurement.
2.	In this paper, the regret bound for the multiobjective loss is T^{1/2}. In the main part and the appendix, the reviewer can only follow the result of the case when m=2, according to Lemma 3. This is a significant issue for this paper since theorem 1 only implies the sub-linear bound when the min-regularized norm method can bound the last term with arbitrary m. Otherwise, the result is a little bit incremental. Another suggestion is that the authors may rewrite the proof to make them easier and more obvious to follow, especially for Corollary 1.


**Summary Of The Paper:**

This paper is about Multi-objective online learning. Compared with the original online learning setting, this paper considers the case that the online algorithm is required to solve multi-sequence input loss functions simultaneously rather than the single input loss function. To measure the performance of the algorithm, this paper involves the sequence-wise PSG (Pareto sub-optimality gap), which extends the regret for the single objective online learning. Meanwhile, this paper involves the min-regularized norm with Multiple Gradient Descent to make a trade-off to the multi-gradient of the loss sequences in the proposed algorithm. Finally, this paper gives a sub-linear “regret” bound to the problem.

**Summary Of The Review:**

It is an interesting work for multi-objective online learning. In this paper, an effective measurement has been involved in online learning for multi-objective functions. Meanwhile, this paper offers an available multi-gradient descent technology in the online version. I would like to re-rate this paper if the presentation and analysis have been improved.

---

### Official Review · Reviewer_GCcZ · 2022-10-27

**Confidence:** 3
**Correctness:** 3
**Technical Novelty And Significance:** 3
**Empirical Novelty And Significance:** 3
**Recommendation:** 8

**Clarity, Quality, Novelty And Reproducibility:**

The paper is very clear, and the quality of the results and presentation is great. The novelty might be somehow limited since it mainly borrows ideas from similar problems in prior literature.

**Strength And Weaknesses:**

**Strengths**
- The paper is very well written and well organized.
- The results are solid, and the authors provide sufficient insights on the issues with alternative solutions.
- There are numerical results that support theoretical results.



**Weaknesses**
- The paper sounds like a nice re-execution of the multi-objective bandits in the full information feedback and online convex optimization. That said, I believe the authors did a great job clarifying the differences and unique challenges in this paper. Hence, I still believe that the contribution of this paper is valuable.
- Is it possible to derive a problem-specific regret lower bound that captures the unique structure of the multi-objective property of the problem? It seems that in the paper, the only remark on the optimality of regret shows that with a proper parameter setting, the result matches the single-objective case. However, it is unclear what is the definition of optimality of regret in the multi-objective setting.

**Summary Of The Paper:**

This paper studies the problem of multi-objective online learning. In the classic online convex optimization problem, there is a single objective function, and the goal is to find the best action that leads to the best value of the objective function in a sequential decision-making setting. This paper extends the problem to a case where there are multiple objective functions. In this setting, then, the notion of best action needs further elaboration to define formally. The reason is that it is possible to have different best actions for different terms of the objective function. In this paper, the notion of Pareto optimality is used to define the Pareto optimal action. In addition, regret in multi-objective settings needs to be carefully redefined.

Then,  after problem formulation, the paper develops an algorithm based on Doubly Regularized Online Mirror Multiple Descent, and the regret of the algorithm is analyzed, and it is shown that it matches the regret in a single objective case.

**Summary Of The Review:**

Overall, I enjoyed reading this paper, and I think it studies a timely and practically relevant problem. The theoretical contribution is nontrivial and nicely executed.

---

### Official Review · Reviewer_WSb3 · 2022-10-27

**Confidence:** 4
**Clarity, Quality, Novelty And Reproducibility:** Globally good
**Correctness:** 3
**Technical Novelty And Significance:** 3
**Empirical Novelty And Significance:** 3
**Recommendation:** 8

**Strength And Weaknesses:**

**Strengths**
- the paper is globally clear and well written
- the topic is of interest to the ICLR community

**Weaknesses**
- I am not fully convinced by the definition of the regret (which is one important contribution of the paper). Typically, in Proposition 1, isn't $\lambda^*$ just selecting the objective with respect to which the regret is the smallest? If the previous regret proposed was indeed too hard, this one seems on the contrary a bit easy. I point out this paper [1], where the authors study multitask Bayesian optimization. In particular in Section 2 they define a notion of regret which is the expectation (over a distribution of directions) of the scalarized regret. This seems to me a sensible definition, as opposed to the best (or the worst) possible direction. Could the authors comment on this point?

**Questions**
- one could imagine a setting where feedback (i.e., the gradient) is received only for a subset of the objectives. How would the regret bound look like in that case?
- does the DR-OMMD achieve improved regret bounds if the objectives satisfy additional regularity assumption (e.g., strong convexity, exp-concavity)? If not, what is the reason?
- it seems to me that the $\lambda$ can be interpreted as learning rates. They are many ways to optimally tune the learning rate and adapt to the comparator, see e.g. [2, Chapter 9]. Could it make sense to use them rather than the regularized min-max norm to derive a sequence of optimal $\lambda_t$?

[1] No-regret Algorithms for Multi-task Bayesian Optimization, Chowdhury and Gopalan 2020
[2] A Modern Introduction to Online Learning, Orabona 2020

**Summary Of The Paper:**

This paper studies multi-objective online learning, a framework in which the learner has to optimize jointly several conflicting objectives. In particular the notion of optimality and discrepancy need to be redefined. The authors propose a definition of the regret based on the sequence-wise Pareto Suboptimality Gap. Then, they propose an algorithm based on the regularized min-norm and prove it achieves a $\sqrt{T}$ regret. Experiments complement the contributions.

**Summary Of The Review:**

Overall, I am not fully convinced by the definition of the regret proposed by the authors. I also proposed some directions to explore to complement the contribution.

---

### Official Review · Reviewer_GxU6 · 2022-10-31

**Confidence:** 4
**Correctness:** 4
**Technical Novelty And Significance:** 4
**Empirical Novelty And Significance:** 4
**Recommendation:** 8

**Clarity, Quality, Novelty And Reproducibility:**

Clarity -- Very well written

Novelty -- Original insights and crisp communication of those insights

Reproducibility -- Yes

**Strength And Weaknesses:**

Strengths -- Very well written paper. The style was nice -- presenting the problem, showing through detailed examples why simple approaches fail and then using those examples to build up to the final algorithm.

Weakness -- The intuition and the effect of \lambda_0 in the algorithm is not clear.  This is however a nit-pick and I believe can easily be fixed in the write-up.

**Summary Of The Paper:**

The paper considers a multi-dimensional version of the online convex optimization. Concretely, the problem studies a sequential interaction between a predictor and an adversarial environment -- at each time, the predictor picks a point in a convex set and the environment subsequently picks M convex functions from the domain to the reals. The goal of the predictor is to make sequential predictions so as to minimize regret with respect to the Pareto frontier of the sum of the M dimensional functions over the time horizon.

The paper contributes to a rigorous definition of regret, provides a variational formula for the same which then aids in analysis. The first observation the paper makes is that directly plugging in the one dimensional OMD to the standard iterative multi-objective MO descent algorithm leads to linear regret. The key observation made here was that the offline algorithm optimizes the mixing weights for the gradients of the M function to the current iterate. However, in the online case, since the environment is adversarial and the regret is with respect to hindsight, converting naively the offline iterative algorithm to an online one yields linear regret. The paper then shows that the simple idea of solving two regularized optimizations -- one to choose the mixing weights and a separate one to choose the next iterate -- yields low regret.

**Summary Of The Review:**

Overall this paper is very well written -- provides a simple algorithm and a new insight (double, separate regularization) for an important practical problem. The style of writing is also very pedagogical and aids the reader in quickly grasping the key concepts.

---

### Decision · Program_Chairs · 2023-01-20

**Decision:**

Accept: notable-top-25%

**Justification For Why Not Higher Score:**

This is not a mainstream ICLR paper. I checked it though and would not mind pushing for an oral if there is a SAC support.

**Justification For Why Not Lower Score:**

This is a solid work (3 reviewers gave it score 8) and it should be highlighted.

**Metareview: Summary, Strengths And Weaknesses:**

This paper studies multi-objective online convex optimization. In each round, the agent chooses a point in a convex set and then the environment reveals an M-dimensional loss function. The agent minimizes regret with respect to the Pareto frontier of the sum of the M-dimensional loss functions over the time horizon.

This work makes solid contributions to the definition of regret, proposes an efficient algorithm for minimizing it, bounds its regret, and demonstrates it empirically. The initial scores of the paper (2x 8 and 2x 5) improved to 3x 8 and 1x 5 after the rebuttal, and show the appreciation of the reviewers for the work. Congratulations to acceptance!

**Note From Pc:**

if the above contains the word "oral" or "spotlight" please see: "oral" presentation means -> notable-top-5% and "spotlight" means -> notable-top-25%. As stated in our emails, we are disassociating presentation type from AC recommendations